# The mechanosensitive ion channel TRAAK is localized to the mammalian node of Ranvier

Stephen G Brohawn[1†‡], Weiwei Wang[1†§], Annie Handler[1], Ernest B Campbell[1], Jürgen R Schwarz[2*], Roderick MacKinnon[1*]

[1]Laboratory of Molecular Neurobiology and Biophysics, Howard Hughes Medical Institute, The Rockefeller University, New York, United States; [2]Institute of Molecular Neurogenetics, Center for Molecular Neurobiology Hamburg (ZMNH), University Medical Center Hamburg-Eppendorf, Hamburg, Germany

*For correspondence:
juergen.schwarz@zmnh.uni-hamburg.de (JRS);
mackinn@mail.rockefeller.edu (RMK)

[†]These authors contributed equally to this work

Present address: [‡]Department of Molecular and Cell Biology, The Helen Wills Neuroscience Institute, University of California, Berkeley, Berkeley, United States; [§]Department of Biophysics, University of Texas Southwestern Medical Center, Dallas, United States

Competing interests: The authors declare that no competing interests exist.

**Abstract** TRAAK is a membrane tension-activated K$^+$ channel that has been associated through behavioral studies to mechanical nociception. We used specific monoclonal antibodies in mice to show that TRAAK is localized exclusively to nodes of Ranvier, the action potential propagating elements of myelinated nerve fibers. Approximately 80 percent of myelinated nerve fibers throughout the central and peripheral nervous system contain TRAAK in what is likely an all-nodes or no-nodes per axon fashion. TRAAK is not observed at the axon initial segment where action potentials are first generated. We used polyclonal antibodies, the TRAAK inhibitor RU2 and node clamp amplifiers to demonstrate the presence and functional properties of TRAAK in rat nerve fibers. TRAAK contributes to the 'leak' K$^+$ current in mammalian nerve fiber conduction by hyperpolarizing the resting membrane potential, thereby increasing Na$^+$ channel availability for action potential propagation. We speculate on why nodes of Ranvier contain a mechanosensitive K$^+$ channel.

DOI: https://doi.org/10.7554/eLife.50403.001

## Introduction

Mechanosensation encompasses the processes by which cells sense physical forces and transduce them into biological responses. Force-gated or mechanosensitive ion channels are cells' fastest force sensors and are gated open by mechanical force to transduce physical stimuli into cellular electrical signals (*Ranade et al., 2015*). First implicated in auditory transduction three decades ago (*Corey and Hudspeth, 1979*), mechanosensitive channels are best known for their roles in the classic senses of touch and hearing. However, mechanosensitive ion channels also underlie less conspicuous force sensations such as proprioception, blood pressure regulation and osmolarity control (*Teng et al., 2015*). The prevalence of mechanically-activated currents in cells has become increasingly clear; nearly all cell types express mechanosensitive channels including stem cells, cancer cells and neurons of the central nervous system, suggesting, perhaps, that force sensation by ion channels may be involved in more aspects of physiology than previously appreciated (*Ranade et al., 2015*; *Teng et al., 2015*; *Anishkin et al., 2014*).

TRAAK (TWIK-related arachidonic acid activated K$^+$ channel) is a mechanosensitive ion channel from the two-pore domain K$^+$-selective (K2P) channel family (*Fink et al., 1998*; *Maingret et al., 1999*). K2P channels give rise to what have traditionally been called 'leak currents' because they approximate the voltage- and time-independent properties of a simple K$^+$-selective hole in the membrane (*Enyedi and Czirják, 2010*). However, it is now appreciated that the activity of K2Ps can be diversely regulated to control cellular excitability (*Renigunta et al., 2015*). Still, the precise

physiological roles of many K2Ps remain poorly understood. TRAAK is gated open by force directly through the lipid bilayer (*Brohawn et al., 2014a*). In the absence of tension, TRAAK displays low activity with an open probability estimated to be less than 1% (*Brohawn et al., 2014b*; *Kang et al., 2005*). The channel is rapidly and increasingly activated from very small applied forces up to the lytic tension of the membrane (*Brohawn et al., 2014b*). Structural work has led to a model for gating and mechanosensitivity of TRAAK to account for these properties (*Brohawn et al., 2014b*; *Brohawn et al., 2012*; *Brohawn, 2015*). Activation of TRAAK involves protein conformational changes that prevent membrane lipids from entering the conduction pathway and blocking K$^+$ passage. TRAAK changes shape upon opening, expanding and becoming less conical and more cylindrical in the plane of the membrane. These shape changes are energetically favored when lateral tension is applied. The consequence is a membrane tension-dependent energy difference between closed and open conformations that favors opening.

These biophysical insights into TRAAK channel function led us to ask how its unique properties are related to its physiological roles. TRAAK is reported to be broadly expressed in neurons of the central and peripheral nervous system, but the resolution of these studies is limited as they investigate the presence of TRAAK transcripts and they differ in detail (*Fink et al., 1998*; *Talley et al., 2001*; *Lein et al., 2007*). TRAAK$^{-/-}$ mice display mechanical and temperature allodynia and enhanced mechanical hyperalgesia during inflammation, consistent with a role for TRAAK in thermal and mechanical nociception (*Noël et al., 2009*). However, whether these phenotypes are due to loss of TRAAK channels within sensory endings, elsewhere in the periphery, or centrally is unknown. Gain-of-function mutations in TRAAK identified in three human families are thought to underlie a complex developmental and neurological disorder FHEIG, an acronym for its characteristic phenotypes of facial dysmorphism, hypertrichosis, epilepsy, intellectual disability, and gingival outgrowth (*Bauer et al., 2018*). This suggests a broader role for TRAAK activity in development and central nervous system function. The current lack of understanding of the precise localization of TRAAK channels precludes a deeper understanding of the biological roles for which the channel has evolved.

Axons of jawed vertebrates contain alternating non-excitable insulated regions where the axonal membrane is wrapped in myelin to increase membrane resistance and decrease capacitance, and excitable regions where the axonal membrane is exposed to enable firing and regeneration of action potentials. Nodes of Ranvier are the periodic ~1 μm gaps in myelination where the action potential is regenerated. Nodes and the immediately surrounding regions under the myelin sheath constitute sharply delineated functional domains with well-defined molecular components (*Rasband and Shrager, 2000*; *Arroyo, 2004*; *Rasband and Peles, 2015*; *Vogel and Schwarz, 1995*). Nodal membranes contain a high density of voltage-gated Na$^+$ channels (Na$_v$1.6), adhesion molecules, and scaffolding components including ankyrin G (AnkG). In addition, K$_v$7.2/K$_V$7.3 (KCNQ2/3) channels are incorporated into the nodal membrane (*Schwarz et al., 2006*; *Devaux et al., 2004*). Flanking the node are paranodes, tight cell-cell junctions between axonal and glial membranes made in part by Contactin-associated protein 1 (Caspr1). Flanking the paranodes are juxtaparanodes, which contain voltage-gated K$^+$ channels (K$_v$1.1 and K$_v$1.2) (*Schwarz et al., 2006*; *Hille, 1967*; *Stämpfli and Hille, 1976*; *Chiu et al., 1979*; *Röper and Schwarz, 1989*; *Chiu and Ritchie, 1981*).

In this study we show that the mechanosensitive TRAAK channel is localized to nodes of Ranvier in myelinated axons throughout the mammalian nervous system. While it has been known for about forty years that the K$^+$ conductance in mammalian nodes is predominantly composed of leak-type rather than voltage-gated channels (*Chiu et al., 1979*; *Röper and Schwarz, 1989*; *Chiu and Ritchie, 1981*; *Brismar and Schwarz, 1985*), TRAAK is, to our knowledge, the first molecularly identified component of this conductance. We demonstrate that the basal activity of TRAAK is involved in maintaining a negative nodal resting potential to increase nodal Na$_v$ channel availability. We further speculate on possible roles for mechanical activation of TRAAK in the nodal membrane.

## Results

### Localization of TRAAK in the nervous system

Our previous X-ray crystallographic studies of *H. sapiens* TRAAK utilized antigen binding fragments (Fabs) of a mouse monoclonal antibody raised against the channel to facilitate crystal packing

(*Brohawn et al., 2014b*; *Brohawn et al., 2013*). The Fabs bound to a structured extracellular epitope of human TRAAK and specifically labeled the channel expressed in cultured cells. We reasoned that analogous antibodies targeting *M. musculus* TRAAK could serve as specific reagents for immunolocalization of the channel in tissue and provide insight into the channel's physiological roles. Monoclonal antibodies were therefore raised in Armenian hamsters (*C. migratorius*) against purified *M. musculus* TRAAK. The functional integrity of purified TRAAK was verified in planar lipid bilayer recordings of reconstituted channels that show K$^+$-selective currents with properties that recapitulate those of TRAAK channels expressed in cells (*Figure 1—figure supplement 1A*). Candidate monoclonal antibodies were screened to identify one that met the following criteria: (i) it was biochemically well behaved (*Figure 1—figure supplement 1B*), (ii) it formed a tight complex with TRAAK (*Figure 1—figure supplement 1C*), (iii) it recognized an extracellular epitope (*Figure 1A*), and (iv) it specifically labeled mouse TRAAK and not related mouse K2P channels (*Figure 1—figure supplement 2*). The utility of this antibody for immunolocalization of TRAAK in mice was confirmed by comparing immunostaining of tissue from *TRAAK$^{+/+}$* and *TRAAK$^{-/-}$* animals (*Figure 1B*).

To further validate and characterize the nature of the antibody-channel interaction, we determined the X-ray crystal structure of antibody Fab fragments in complex with mouse TRAAK to 2.8 Å resolution. The structure was solved by molecular replacement using Fab and channel search models and refined to a final R$_{work}$ = 0.25 and R$_{free}$ = 0.29 (*Figure 1C*, *Supplementary file 1*). The structure consists of one channel with two bound Fabs on opposite sides of the channel extracellular helical cap in the asymmetric unit. Consistent with a strong interaction, the Fabs bind to large epitopes (1003 Å$^2$ average buried surface area) with high shape complementarity. The antibody binding site spans both subunits of the dimeric channel, with cap helix 1 of one protomer forming ~2/3 of the surface and cap helix two from the second protomer forming the remaining ~1/3. Most of the twenty

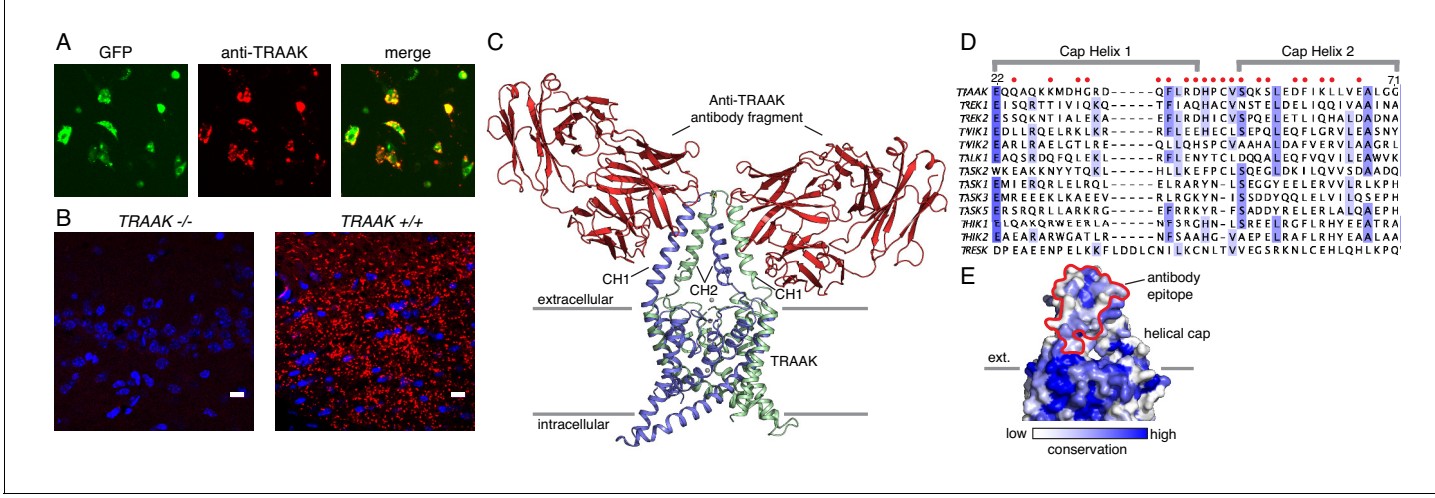

**Figure 1.** A specific anti-mouse TRAAK antibody targets the channel extracellular cap. (**A**) GFP fluorescence (left, green), anti-TRAAK antibody labeling (center, red), and merged image (right) from a field of live CHO-K1 cells expressing mouse TRAAK-EGFP. The anti-TRAAK antibody labels live cells expressing TRAAK indicative of extracellular epitope recognition. (**B**) Anti-TRAAK antibody labeling (red) and DAPI (blue) in brain sections from *TRAAK$^{-/-}$* and *TRAAK$^{+/+}$* animals. Scale bar = 10 μm. (**C**) TRAAK-Fab complex structure viewed from the membrane plane with extracellular solution above. Antibody Fab fragments (red) bind the extracellular cap of TRAAK (shown with one chain of the dimeric channel colored blue and the second green). Potassium ions are shown as gray spheres. Approximate boundaries of the membrane bilayer are indicated with gray bars. Cap helices 1 and 2 are indicated (CH1, CH2). (**D**) Sequence alignment of the helical cap region of mouse K2P channels (amino acid numbers correspond to mouse TRAAK). Residues of mouse TRAAK that form the antibody binding epitope are indicated with red circles above the sequence. (**E**) Surface view of the helical cap from the membrane plane colored according to conservation among mouse K2Ps as in (**D**). Antibody binding surface is outlined in red.

DOI: https://doi.org/10.7554/eLife.50403.002

The following figure supplements are available for figure 1:

**Figure supplement 1.** Anti-TRAAK antibodies form biochemically stable channel complexes without impacting channel function.
DOI: https://doi.org/10.7554/eLife.50403.003
**Figure supplement 2.** Anti-TRAAK monoclonal antibody selectively labels TRAAK among all functionally expressed mouse K2P channels.
DOI: https://doi.org/10.7554/eLife.50403.004

epitope-forming residues are poorly conserved among K2P channels (*Figure 1D*), offering a molecular explanation for the high specificity of the antibody for TRAAK among all mouse K2Ps. Some K2Ps can heterodimerize to form functional channels (*Levitz et al., 2016*; *Blin et al., 2016*; *Lengyel et al., 2016*). TRAAK heterodimers, if present in vivo, are not expected to interact strongly with the antibody as the recognized epitope includes TRAAK-specific residues contributed by each protomer.

$TRAAK^{-/-}$ mice show defects in mechanical and thermal nociception relative to wild-type animals (*Noël et al., 2009*) and the presence of TRAAK in somatosensory endings has been proposed to account for these differences. However, we found no evidence for channel localization in sensory structures including lanceolate endings surrounding hair follicles and Merkel cells within touch domes of glabrous skin (*Figure 2—figure supplement 1A,B*). Instead, TRAAK-labeling was restricted to ~1 um$^3$ puncta throughout the nervous system (e.g. *Figure 1B*). These sites are not synapses as TRAAK staining is spatially distinct from PSD95-labeled terminals (*Figure 2—figure supplement 1C*). Consistent with previous reports of neuronally restricted TRAAK expression (*Fink et al., 1998*), antibody staining was distinct from astrocytic (GFAP-positive) and microglial (IBA1-positive) processes (*Figure 2—figure supplement 1D,E*).

We found TRAAK immunolabeling exclusively at nodes of Ranvier throughout the nervous system. *Figure 2A* shows a representative section of hindbrain white matter enriched in myelinated axons from a $Thy1^{EGFP}$ animal that expresses EGFP within a sparse subset of neurons. The close superposition of TRAAK and EGFP labeling is consistent with neuronal expression of the channel (*Figure 2A*). Within neurons, TRAAK is flanked by Caspr1, a hallmark constituent of paranodal regions flanking nodes of Ranvier (*Rasband and Shrager, 2000*; *Arroyo, 2004*; *Rasband and Peles, 2015*). Co-staining with additional markers confirmed that TRAAK localized to nodes of Ranvier: TRAAK co-localizes with known nodal components Na$_v$1.6 and AnkG in sites delineated by flanking Caspr1 staining (*Figure 2B,D*) and TRAAK, Caspr1, and K$_v$1.2 are present in three adjacent and non-overlapping domains (*Figure 2C*) corresponding to a central TRAAK-containing node, flanking Caspr1-containing paranodes, and K$_v$1.2-containing juxtaparanodes (*Figure 2E*).

Action potentials originate in axon initial segments prior to being propagated to sequential nodes of Ranvier during saltatory conduction. These two electrically excitable domains of the axon contain a common repertoire of scaffolding proteins (including AnkG and β4-spectrin) and ion channels (including Na$_v$1.1, Na$_v$1.2, Na$_v$1.6, K$_v$7.2 and K$_v$7.3) (*Rasband and Peles, 2015*; *Normand and Rasband, 2015*; *Arancibia-Carcamo and Attwell, 2014*). We asked whether TRAAK is similarly localized to axon initial segments in addition to nodes of Ranvier. *Figure 2D* shows a region of cortex in which both nodes of Ranvier and initial segments are present and labeled by AnkG. While co-localization of TRAAK and AnkG is observed in nodes of Ranvier, TRAAK-labeling is absent from initial segments. TRAAK appears to be restricted to the nodal compartment.

We next assessed localization of TRAAK outside of the brain and found that TRAAK is widely distributed throughout the nervous system. TRAAK is found in nodes of Ranvier in optic nerve, spinal cord, and sciatic nerve co-localized with AnkG and flanked by paranodal Caspr1 and juxtaparanodal Kv1.2 (*Figure 3A–C*). These data establish TRAAK as an ion channel component of nodes of Ranvier in myelinated axons throughout the nervous system.

TRAAK labeling was not identified in every node of Ranvier. To characterize differences between nodes with and without TRAAK, the percentage of nodes containing TRAAK in different regions of the nervous system was quantified by identifying nodes flanked on both sides by paranodal Caspr1 and scoring for the presence of anti-TRAAK immunofluorescence (*Figure 3—figure supplement 1*). Nodes in most regions examined were predominantly and comparably TRAAK-positive. For example, 73 ± 5% of nodes in the hippocampal-proximal fiber tracts of the fornix and fimbria contain TRAAK (*Figure 3—figure supplement 1A,H*). A similar percentage (79 ± 4%) of nodes in the optic nerve are TRAAK-positive (*Figure 3A*, *Figure 3—figure supplement 1H*). However, in one brain region we identified fiber bundles in which nodes were predominantly TRAAK-negative: (*Figure 3—figure supplement 1B,C,H*): 16 ± 8% of nodes in fields of the ventral striatum and internal capsule containing these tracts were labeled with anti-TRAAK antibody.

One possible explanation for these results is that TRAAK is present within particular classes of neurons and excluded from others. We took advantage of the spatial segregation of different neuronal types in the PNS to determine whether TRAAK is specifically expressed in motor or sensory neurons. Motor neuron enriched ventral horns, sensory neuron enriched dorsal horns, and regions of the projected nerve containing a mixed population of motor and sensory fibers contain comparably high

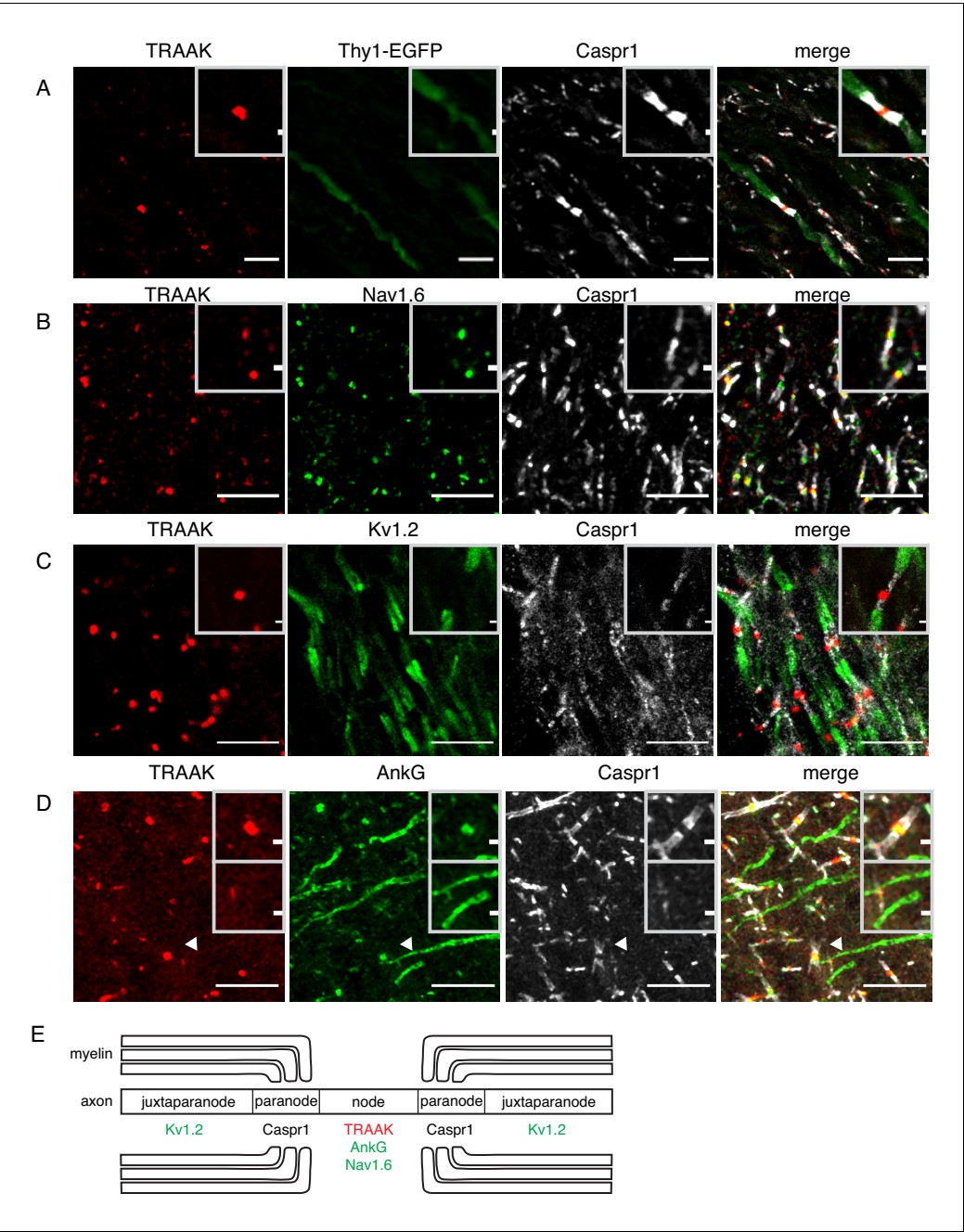

**Figure 2.** TRAAK is localized to nodes of Ranvier in the central nervous system. (**A**) Immunostaining of TRAAK (red), Thy1-EGFP (sparsely labeled neurons, green) and Caspr1 (present in paranodes, white) in a hindbrain section. (**B–D**) Immunostaining of TRAAK (red), Caspr1 (white) and (**B**) Na$_v$1.6 (present in nodes of Ranvier), (**C**) K$_v$1.2 (present in juxtaparanodes), and (**D**) AnkG (present in nodes of Ranvier and axon initial segments, one of which is labeled with an arrowhead) in the (**A–C**) hippocampal-proximal fiber tracts of the fornix or (**D**) cortex. Scale bars = 10 μm. Insets show a magnified view of (**A–D**) a single node or (**D**, lower inset) an axon intial segment. Inset scale bars = 1 μm. (**E**) Cartoon representation of immunolocalization data. TRAAK, Nav1.6, and AnkG are localized to nodes of Ranvier in myelinated axons which are flanked by Caspr1 in paranodes and K$_v$1.2 in juxtaparanodes.

DOI: https://doi.org/10.7554/eLife.50403.005

The following figure supplement is available for figure 2:

**Figure supplement 1.** TRAAK is not found in sensory endings, synapses, astrocytes or microglia.

DOI: https://doi.org/10.7554/eLife.50403.006

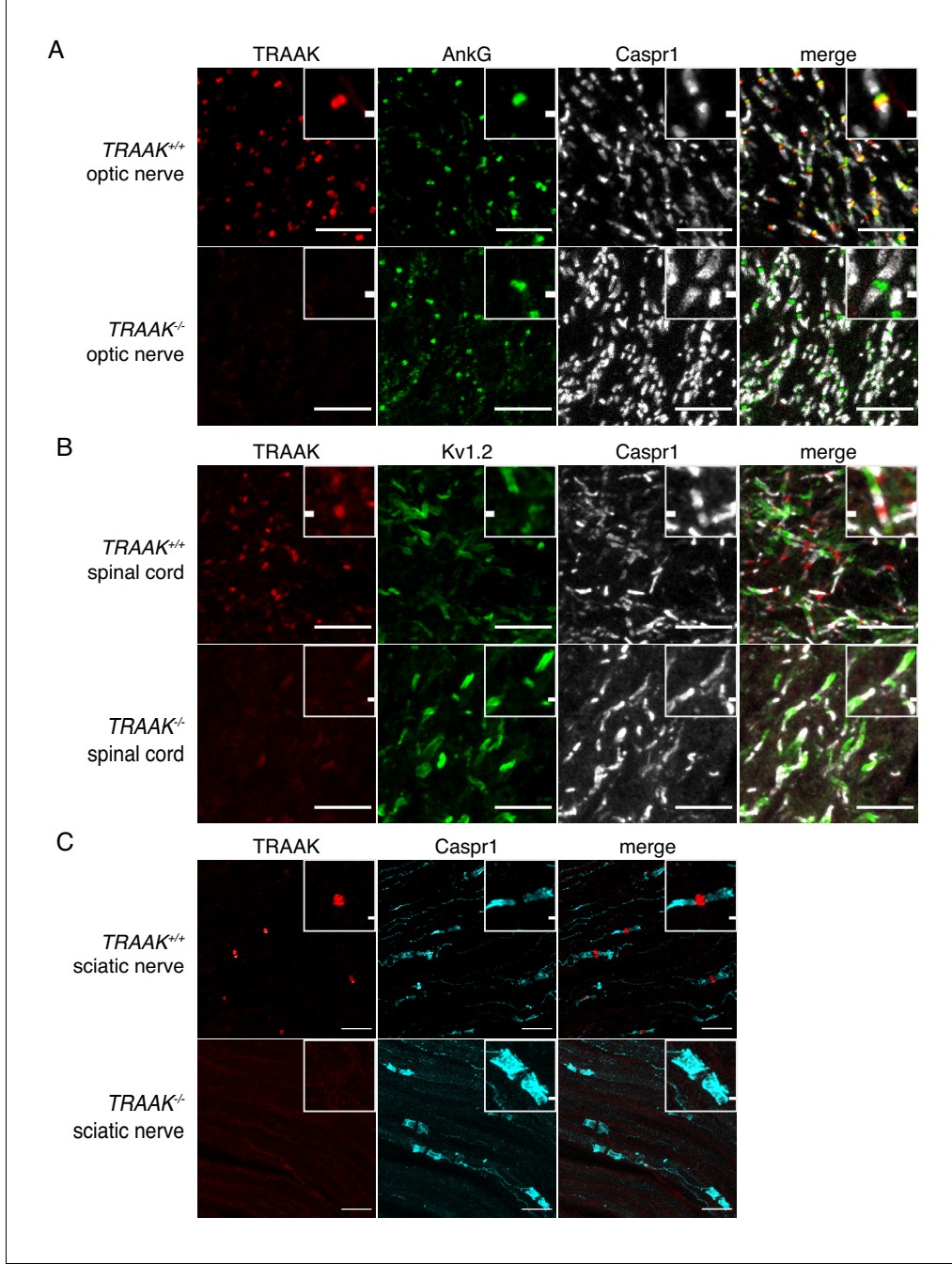

**Figure 3.** TRAAK is localized to nodes of Ranvier in the peripheral nervous system. (**A**) Immunostaining of TRAAK (red), AnkG (green) and Caspr1 (white) in optic nerve from *TRAAK*<sup>+/+</sup> (upper row) and *TRAAK*<sup>-/-</sup> (lower row) animals. (**B**) Immunostaining of TRAAK (red), $K_v$1.2 (green) and Caspr1 (white) in spinal cord from *TRAAK*<sup>+/+</sup> and *TRAAK*<sup>-/-</sup> animals. (**C**) Immunostaining of TRAAK (red) and Caspr1 (cyan) in sciatic nerve from *TRAAK*<sup>+/+</sup> and *TRAAK*<sup>-/-</sup> animals. Scale bars = 10 µm. Insets show a magnified view of a single node. Inset scale bars = 1 µm.
DOI: https://doi.org/10.7554/eLife.50403.007

The following figure supplement is available for figure 3:

**Figure supplement 1.** Quantification of TRAAK nodal localization.
DOI: https://doi.org/10.7554/eLife.50403.008

proportions of TRAAK-positive nodes (82 ± 4%, 86 ± 0.5% and 82 ± 0.2% respectively (mean ± standard deviation, n = 3)) (*Figure 3—figure supplement 1D–H*). Other regions containing mixed populations of neuronal types gave similar results. 81 ± 2% of nodes in the spinal cord contained TRAAK (*Figure 3B*, *Figure 3—figure supplement 1G*) and no apparent difference was observed between motor and sensory tracts. In the sciatic nerve, 82 ± 7% of nodes are TRAAK-positive (*Figure 3C*, *Figure 3—figure supplement 1G*). We conclude that TRAAK is not specifically localized to either sensory or motor fibers and is present in a majority of nodes in the PNS.

It is impossible from sectioned tissue to determine whether TRAAK-positive and TRAAK-negative nodes can both occur within a single nerve fiber. To query this point, we isolated five axons from sciatic nerve; four of these contained three nodes and one contained two (*Figure 4A,B*). We observe that all nodes within a fiber are either positive (axons 1–3) or negative (axons 4–5). If we assume that positive and negative nodes can exist in a single fiber with random occurrence then, given that 80% of nodes in the sciatic nerve are positive (*Figure 3—figure supplement 1*), the probability of not observing a 'mixed' fiber among the five dissected nerves is $((0.8)^3 + (0.2)^3)^4((0.8)^2 + (0.2)^2)$, which is

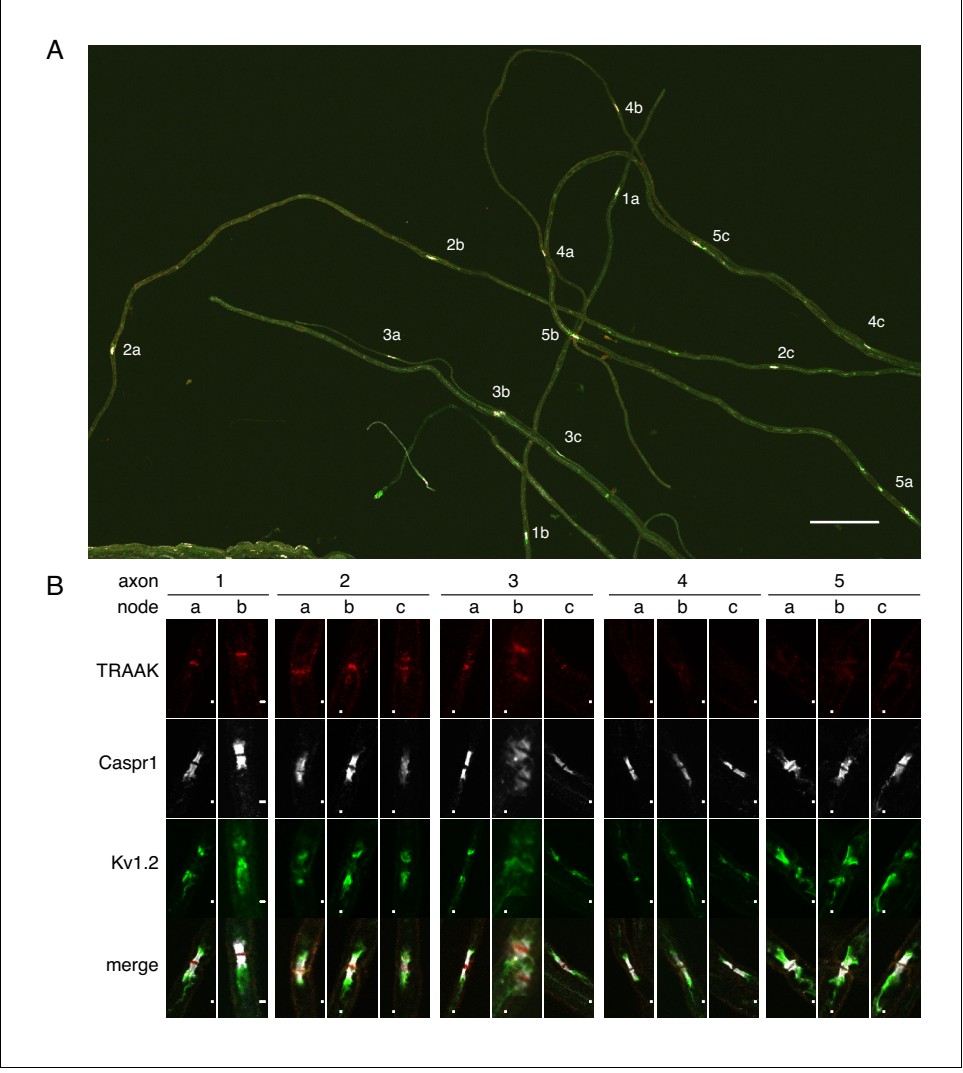

**Figure 4.** All or none distribution of TRAAK in nodes within single axons. (A) Immunostaining of TRAAK (red), Caspr1 (white) and $K_v1.2$ (green) in a field of teased axons from sciatic nerve. Scale bar = 10 μm. Individual axons with multiple node-internode segments in the field are numbered (1-5) with corresponding nodes denoted by letters (a–c). (B) Magnified view of each node marked (A). Scale bar = 1 μm. Each axon has either all TRAAK-positive or all TRAAK-negative nodes.
DOI: https://doi.org/10.7554/eLife.50403.009

slightly less than 0.05. The occurrence of TRAAK in nodes is therefore unlikely to be random. The data are more consistent with there being two kinds of fibers, those with and those without TRAAK.

## Measurements of TRAAK currents in the node of Ranvier

To analyze the physiological function of TRAAK channels in the node of Ranvier we performed electrophysiological recordings from single myelinated axons using a voltage clamp technique known as node clamp (*Nonner, 1969*). We first attempted recordings from mouse axons, but were unsuccessful due to the difficultly in dissecting, isolating, and transferring single fibers from the mouse sciatic nerve to the recording chamber without damage. We thus turned to species from which successful recordings have been made, frog (*Xenopus laveis*) and rat (*Rattus norvegicus*) (*Schwarz and Eikhof, 1987*; *Neumcke et al., 1987*). In each case, we isolated single nerve fibers and recorded from single nodes of Ranvier (*Figure 5A–D*). These measurements were made using one of the few remaining functional amplifiers designed by *Nonner (1969)* and using a new version we designed using operational amplifiers (*Figure 5—figure supplements 1,2*). Both node clamp

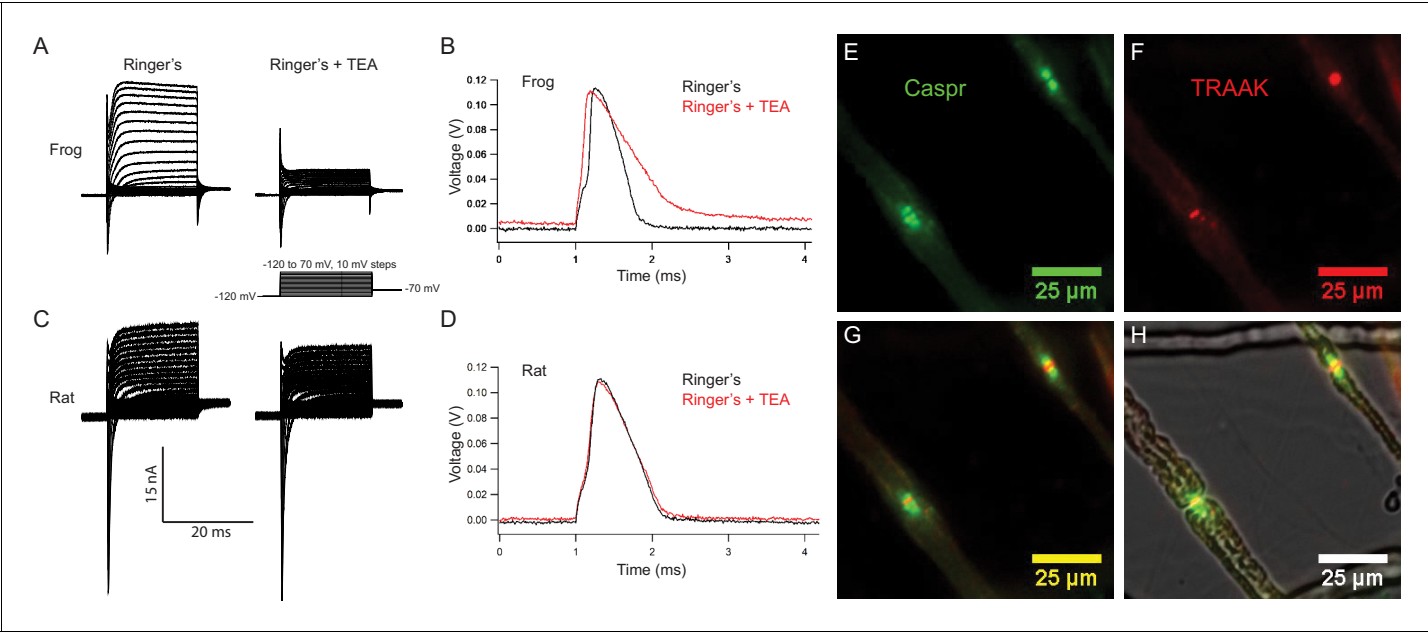

**Figure 5.** Rat nodes of Ranvier express TRAAK and are insensitive to the voltage-gated K$^+$ channel blocker TEA. **(A–D)** Representative voltage clamp and current clamp recordings from isolated single frog and rat nodes of Ranvier. **(A)** Voltage steps recorded in a typical frog node without (left) and with (right) 10 mM TEA. **(B)** Action potentials in a frog node elicited by current injection in the absence (black) and the presence (red) of TEA. Note that the duration of action potential became longer in the presence of TEA (Frog Ringer's amplitude = 114.5 ± 1.2 mV, duration = 0.45 ± 0.006 ms (at 50% amplitude); Frog Ringers + TEA depolarization = 4.1 ± 1.7 mV, amplitude = 110.3 ± 1.3 mV, duration = 0.75 ± 0.06 ms; amplitude reduction with TEA = 4.2 ± 1.8 mV; n = 4). **(C)** Voltage steps recorded in a typical rat node without (left) and with (right) 10 mM TEA. **(D)** Action potentials in a rat node elicited by current injection in the absence (black) and the presence (red) of TEA. The duration of action potential did not change by the application of TEA. (Rat Ringer's amplitude = 108.4 ± 2.5 mV, duration = 0.55 ± 0.02 ms (at 50% amplitude); Rat Ringers + TEA depolarization = 3.7 ± 1.0 mV, amplitude = 103.7 ± 1.6, duration = 0.61 ± 0.02 ms; amplitude reduction with TEA = 4.7 ± 3.0 mV; n = 7). Recordings were performed at room temperature. **(E)** Immunofluorescence of the paranodal protein CASPR (green) in dissected rat sciatic nerve fibers. **(F)** Immunofluorescence of TRAAK (red) in the same nerve fibers. **(G)** Overlay of **(E)** and **(F)**. **(H)** Overlay of the fluorescence image in **(G)** with bright-field image.
DOI: https://doi.org/10.7554/eLife.50403.010

The following figure supplements are available for figure 5:

**Figure supplement 1.** Nerve chamber and schematic of voltage clamping the node of Ranvier.
DOI: https://doi.org/10.7554/eLife.50403.011

**Figure supplement 2.** Schematic of the Node clamp based on operational amplifiers.
DOI: https://doi.org/10.7554/eLife.50403.012

**Figure supplement 3.** Rabbit polyclonal antibody 2134 cross-reacts with rat TRAAK.
DOI: https://doi.org/10.7554/eLife.50403.013

devices yielded similar results. All recordings described below were made from thick myelinated fibers with diameters of 10–15 μm like those shown in *Figure 5E–H*.

Representative membrane currents recorded from single frog and rat nodes of Ranvier under voltage clamp are compared in *Figure 5A,C*. $Na^+$ and $K^+$ currents were elicited with depolarizing potential steps of increasing amplitude preceded by a 50 ms potential step to −120 mV to remove steady-state $Na_V$ channel inactivation. In nodes of both species, early transient $Na^+$ currents and delayed outward $K^+$ currents were recorded. Notably, application of the voltage-gated $K^+$ channel blocker TEA (10 mM) revealed clear differences between the $K^+$ currents in frog and rat nodes. In frog nodes,~75% of the outward $K^+$ current was blocked by TEA, whereas the remaining TEA-insensitive current is predominantly mediated by leak-type current (*Hille, 1967*; *Stämpfli and Hille, 1976*). In contrast, in rat nodes, the majority of the $K^+$ current is TEA-insensitive and is largely composed of leak-type current (*Chiu et al., 1979*; *Chiu and Ritchie, 1981*; *Brismar, 1980*). These differences in outward $K^+$ current likely reflect the difference in $K_V$ channel distribution in rat and frog nodes of Ranvier. In the frog, TEA-sensitive $K_V$ channels are co-localized with $Na_V1.6$ channels at the node (*Hille, 1967*), while in the rat, $K_V1$ channels are restricted to the juxtaparanodal membrane and are absent in nodal axolemma uncovered by Schwann cell loops (*Chiu et al., 1979*; *Röper and Schwarz, 1989*; *Brismar, 1980*). Therefore, $K_V1$ channels likely do not contribute to the nodal outward current in rat and any TEA-sensitive outward current in the rat node is likely mediated by $K_V7.2/7.3$ channels that co-localize with nodal $Na_V$ channels (*Schwarz et al., 2006*; *Devaux et al., 2004*). As expected, $Na^+$ currents are unaffected by TEA and are outwardly directed at potential steps larger than the $Na^+$ equilibrium potential (*Figure 5A,C*).

*Figure 5B and D* demonstrate the effect of TEA on the resting potential and action potential from a frog and a rat node of Ranvier, respectively. Frog action potentials had an amplitude of ~115 mV and sometimes exhibit a noticeable afterhyperpolarization. Application of TEA depolarized the resting potential by ~4 mV, reduced the action potential amplitude by ~4 mV, prolonged the duration of the action potential by a factor of almost two, and typically abolished the afterhyperpolarization. In contrast, action potentials recorded from rat nodes showed no afterhyperpolarization. Application of TEA depolarized the rat node resting potential by ~4 mV and reduced the amplitude of the action potential by ~5 mV, but the duration of the action potentials remained almost unchanged. Taken together, these data show that repolarization of the frog action potential is brought about by a combination of $Na_V$ channel inactivation and delayed activation of TEA-sensitive $K_V$ channels (*Frankenhaeuser and Huxley, 1964*), very similar to the ionic mechanism underlying action potential repolarization in the squid giant axon (*Hodgkin and Huxley, 1952*). In contrast, TEA-sensitive Kv channels do not play a significant role in repolarization of the action potential of the rat node of Ranvier (*Chiu et al., 1979*; *Chiu and Ritchie, 1981*; *Brismar, 1980*). In the mammalian node, $K^+$-selective leak currents provide the $K^+$ conductance necessary for node repolarization and the time course of action potential repolarization is determined by that of $Na^+$ channel inactivation (*Schwarz et al., 1995*).

Does TRAAK contribute to the leak-type currents in rat nodes? We first asked whether localization of TRAAK in rat is similar to mouse. We generated a polyclonal antibody against mouse TRAAK that cross-reacts with the rat channel (*Figure 5—figure supplement 3*) and using this antibody, found that rat TRAAK is similarly localized to nodes of Ranvier (*Figure 5E–H*). We next took a pharmacological approach to assess whether TRAAK contributes to nodal currents in rat axons. The small molecule RU2 was identified in a high-throughput $K^+$ channel drug screen as a mouse TRAAK inhibitor (*Su et al., 2016*). We verified that RU2 also blocks rat TRAAK heterologously expressed in CHO cells (*Figure 6—figure supplement 1*). The RU2-sensitive current appears as a $K^+$ leak current with rectification. In nodes of Ranvier recorded at 32-37° C, 10 or 20 μM RU2 blocked about 20% of the outward $K^+$ current (*Figure 6A–D*, n = 16 RU2-sensitive nodes). We note that the RU2-sensitive currents in rat nodes have different voltage- and time-dependent opening rates compared to the TRAAK currents in CHO cells (*Figure 6D,E*; *Figure 6—figure supplement 1*). This difference in gating is consistent with the previously demonstrated behavior of TRAAK channels at different levels of activation (*Schewe et al., 2016*). The currents in CHO cells are consistent with high levels of TRAAK activation and those in nodes of Ranvier are consistent with sub-maximally activated TRAAK. RU2 application did not affect the $Na^+$ current amplitude, only an outward $K^+$ current in the absence (*Figure 6A*) and presence (*Figure 6B*) of the $Na^+$ channel inhibitor tetrodotoxin (TTX). Upon removal, the effect of RU2 on the outward $K^+$ current was reversible (n = 8). RU2-sensitive current

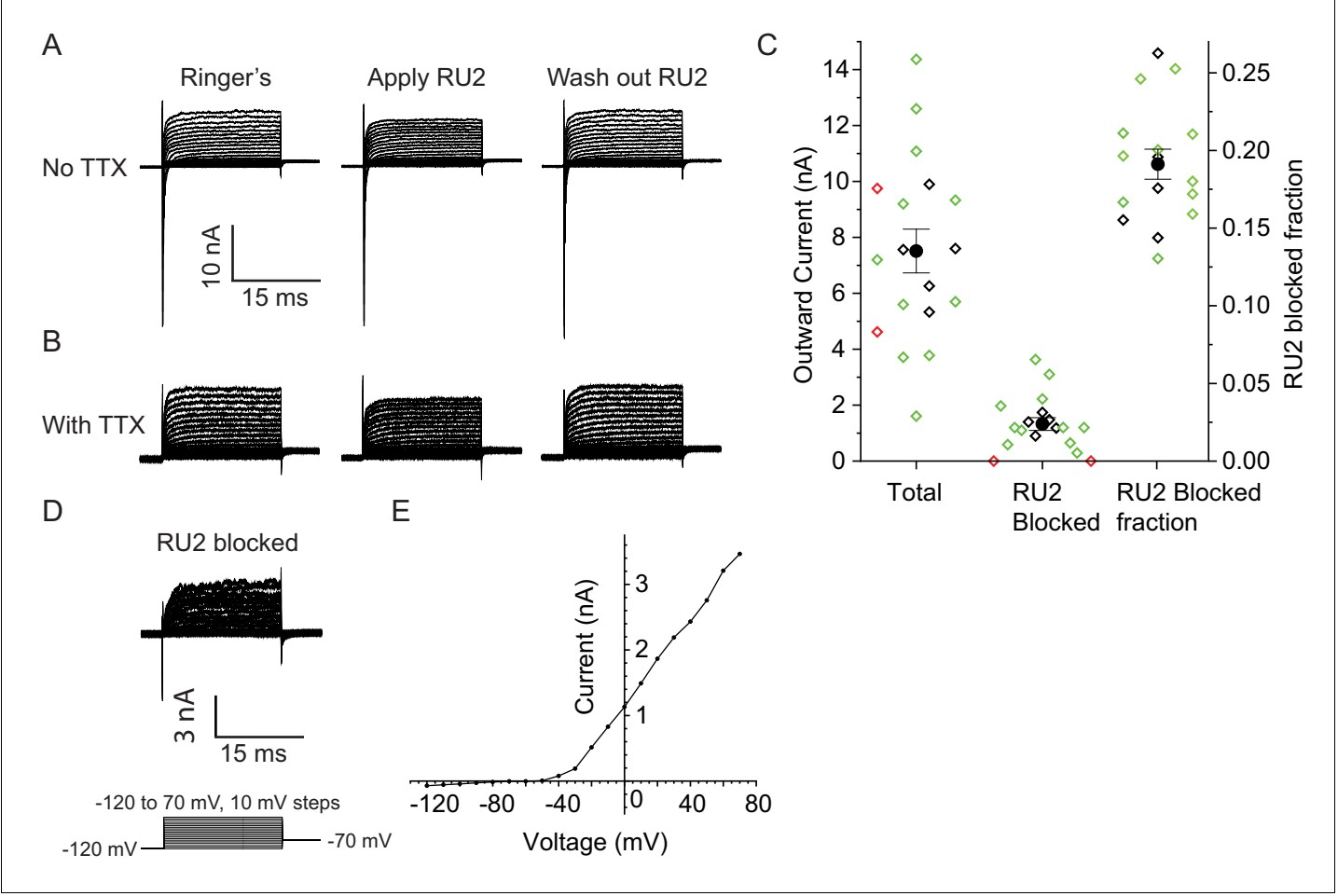

**Figure 6.** RU2 blocked current in the nodes of Ranvier in rat sciatic nerve.  Recordings were performed at 32–37°C. (**A**) Currents recorded during voltage steps applied to a rat nodal membrane in the absence of RU2 (left), in the presence of 20 µM RU2 (middle) and after RU2 wash out (right). (**B**) Same experiments as in panel A but performed in the presence of the Na⁺ channel blocker TTX. (**C**) Quantification of the total and RU2-sensitive nodal current at 50 mV from 18 nodes recorded at 32–37°C (green: no TTX, black: with TTX). 2 out of the 18 nodes did not have RU2-sensitive current (red) and were not included in the statistics. The total outward current was $8 \pm 1$ nA, the RU2 sensitive current was $1.5 \pm 0.3$ nA and the fraction of RU2 sensitive current in the total current was $0.19 \pm 0.01$ (mean ± S.E.M.). (**D**) RU2 blocked current calculated from subtracting voltage steps in the presence of RU2 (panel B middle) from that in the absence of RU2 (panel B left). (**E**) I-V curve of the RU2 blocked current. Data points are represented as filled circles and connected with straight line segments.

DOI: https://doi.org/10.7554/eLife.50403.014

The following figure supplement is available for figure 6:

**Figure supplement 1.** RU2 blocked current in CHO cells heterologously expressing rat TRAAK.

DOI: https://doi.org/10.7554/eLife.50403.015

was observed in 89% of rat nodes recorded at 32-37° C (**Figure 6C**, n = 18) or 86% of rat nodes recorded at either 32–37 ° C or room temperature (n = 64 nodes). These electrophysiological data are consistent with about 80% of nodes in sciatic nerve being TRAAK-positive by immunolabeling in mice (**Figure 3—figure supplement 1H**). RU2 application had no effect on 4 out of 4 frog nodes.

One consequence of blocking nodal TRAAK on the action potential in rat axons is illustrated in **Figure 7**. Application of RU2 depolarized the nodal membrane by ~4 mV and resulted in a significant reduction in action potential amplitude (**Figure 7A**). This reduction was completely reversible upon washout. Repetitive 100 Hz stimulation exaggerates this effect: in the presence of RU2 the amplitudes of the repetitively elicited action potentials decrease until they reach a steady state level of diminished amplitude (**Figure 7E**). The RU2-induced reduction of the action potential amplitude is likely due to a depolarization-induced decrease in the number of Na$_V$ channels available for

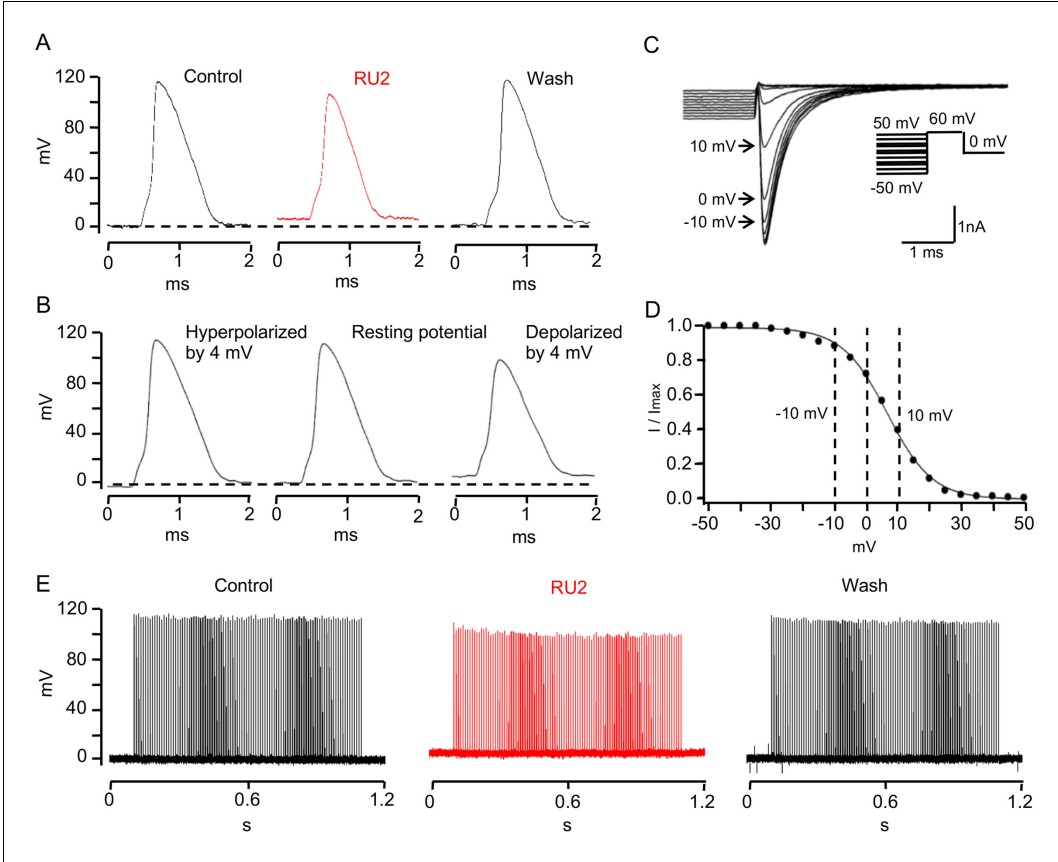

**Figure 7.** TRAAK stabilizes the resting potential of the rat node of Ranvier. (**A**) Representative action potential recorded from a rat node of Ranvier before drug application (left), during application of 10 uM RU2 (red, center), and after drug wash out (right). Application of RU2 depolarized the resting potential by 4 mV and reduced the action potential amplitude. These effects could be washed out. Averaged results: control action potential amplitude = 112.1 ± 5.2 mV (n = 7); with RU2 depolarization of resting potential = 3.8 ± 0.2 mV, action potential amplitude = 98.3 ± 6.7 mV (n = 7); wash-out: repolarization of resting potential = 5.3 ± 1.0 mV, action potential amplitude = 118.3 ± 3.7 mV (n = 3). (**B**) Action potential recorded from a rat node of Ranvier with the resting potential hyperpolarized by 4 mV (left), at the resting potential (center) and depolarized by 4 mV (right). Hyperpolarization increased and depolarization reduced the action potential amplitude. (**C**) Membrane currents used to measure steady-state Na$_V$ inactivation. A 60 mV test pulse elicited a transient inward Na current (inset: pulse protocol). 50 ms depolarizing prepulses of increasing amplitude decreased the peak Na current amplitudes, hyperpolarizing prepulses of increasing amplitudes increased the Na current amplitude until a steady amplitude was obtained. (**D**) Normalized peak Na current amplitudes of the experiment shown in (**C**) plotted versus prepulse potential. (**E**) Repetitive stimulation (100 Hz for 1 s, 0.2 ms just suprathreshold current stimuli) of rat node of Ranvier in Ringer´s solution (control), in the presence of RU2 and after wash-out of RU2.
DOI: https://doi.org/10.7554/eLife.50403.016

activation upon a depolarizing stimulus. This effect is illustrated by the Na$_V$ inactivation curve (**Figure 7C,D**). Here, the resting potential is defined as V = 0 mV (see Materials and methods). Since the inactivation curve is steep near the resting potential, even a small depolarization of 4 mV produces a relatively large increase in the number of inactivated Na$_V$ channels. This results in a reduction of the action potential amplitude similar to that caused by the RU2-induced depolarization. These experiments demonstrate that TRAAK-mediated leakage current is important for the stabilization of a negative resting potential and maximum spike amplitude. The same function has been ascribed to the resting conductance of the nodal M-current mediated by Kv7 channels (**Battefeld et al., 2014**).

## Discussion

We have found that the mechanosensitive K2P TRAAK is localized to nodes of Ranvier in mammalian myelinated axons. Our imaging data are consistent with TRAAK being exclusively localized to nodes of Ranvier as we did not detect TRAAK elsewhere in the nervous system, including within initial segments of myelinated axons that otherwise share a common set of ion channels and scaffolding components. This finding implies that unlike other nodal channels (including Na$_v$s and K$_v$3.1, 7.2, and 7.3 channels) (*Rasband and Peles, 2015*; *Arancibia-Carcamo and Attwell, 2014*), TRAAK localization to nodes is not mediated through a direct interaction with AnkG. TRAAK is found in most (~80%), but not all, nodes we imaged and from experiments with teased fibers we conclude that neurons likely either have TRAAK in all nodes or no nodes within their axons. It may be that the presence or absence of TRAAK in nodes gives rise to functional differences between neurons, perhaps in action potential shape or firing properties. We were not able to further distinguish TRAAK-positive and –negative neurons, so evaluating potential differences between neurons with and without TRAAK will require additional genetic, anatomical and electrophysiological characterization of neuronal classes.

Our data do not exclude the possibility that other K2P channels are also present in the node of Ranvier. RU2 is not entirely specific for TRAAK among K2P channels and therefore some of the RU2-sensitive component could be due to TRESK or TASK3, which are also RU2-sensitive (*Su et al., 2016*). However, that the fraction of TRAAK-positive nodes by immunohistochemistry is about equal to the RU2-sensitive fraction by electrophysiology is an observation best explained (although not 'proved') if TRAAK is the only RU2-sensitive K2P channel in the node. Since TRAAK accounts for only about 20% of the total leak current under our recording conditions, we think it is likely that other RU2-insensitive K2P channels are present in nodes. Their identification is not the focus of this study.

TRAAK channels are only present in organisms that have myelinated axons and nodes of Ranvier – the gnathostomes or jawed vertebrates for which evolution of axonal myelination with myelin basic protein is a defining characteristic (*Nawaz et al., 2013*; *Nave and Werner, 2014*). It is clear from the comparison of recordings from single rat and frog nodes presented here that the contribution of TRAAK to nodal currents is not strictly conserved between species with TRAAK. In frog nodes, K$_v$ currents play the dominant role in repolarizing the action potential. In rat nodes, however, K$_v$ channels make a small contribution and TRAAK is a component of the leak-type outward current. The basal activity of TRAAK plays a role in maintaining a negative resting potential, thereby increasing Na$_v$ channel availability and maximizing the action potential amplitude in rats. The presence of TRAAK in mammalian nodes raises additional questions. Are the unique functional properties of TRAAK tuned to specific physiological aspects of node of Ranvier? What might other roles of TRAAK in nodes be? We speculate on three possibilities below.

First, the basal activity of TRAAK contributes to repolarizing the nodal membrane during a spike. Indeed, mammalian myelinated axons generally lack K$_v$ channel activity that could repolarize the membrane. In mammals, among axonal K$_v$ channels, K$_v$1 channels are juxtaparanodally restricted and do not normally influence action potential repolarization (*Arroyo, 2004*; *Schwarz et al., 2006*; *Röper and Schwarz, 1989*; *Chiu and Ritchie, 1981*; *Schwarz et al., 1995*), K$_v$3 channels present in central nodes of Ranvier (*Devaux et al., 2003*) play minor roles after development of mature nodes, and slowly opening K$_v$7 channels are thought to contribute mostly to the resting membrane potential and thus Na$_v$ channel availability (*Schwarz et al., 2006*). As our recordings presented here show, repolarization of the mammalian node depends on a K$^+$ 'leak' conductance: a K$^+$-selective, approximately voltage-independent, TEA-insensitive K$^+$ channel. These properties are hallmarks of K2P channel activity (*Enyedi and Czirják, 2010*). We demonstrate that TRAAK contributes to this leak K$^+$ conductance. The low open probability of TRAAK in resting membranes in the absence of mechanical force (<1% open) may be sufficient to support nodal repolarization. K2Ps are capable (in simulations and culture cells) of supporting repolarization and repetitive spiking in the absence of K$_v$ activity (*MacKenzie et al., 2015*). It may be that utilization of basally open TRAAK channels to repolarize nodes enables higher frequency axonal firing compared to utilization of slower-opening K$_v$ channels.

Second, TRAAK could serve a mechanoprotective role in nodes of Ranvier. Mechanical insult can stretch and injure nerves and elaborate anatomical features exist to protect nerves and their enclosed axons from such injury at the expense of increased axonal length and action potential transmission time. Nerves in rorqual whale tongues offer an extreme example by allowing extension

more than twice their unstretched length without damage; this property accommodates the tremendous volume of water engulfed during feeding (*Vogl et al., 2015*). In fact, most nerves contain bundles of axons that are pleated like an accordion along the long axis of a nerve such that they can unfold during stretch to prevent mechanical damage (*Topp and Boyd, 2006*). Within nerves, nodes appear to be particularly mechanically vulnerable sites. Nerve stretch could generate sufficient membrane tension to open depolarizing channels, perhaps weakly mechanosensitive $Na_v$ channels (*Tabarean et al., 1999*). Unchecked depolarization could initiate ectopic action potentials mid-axon and destroy the fidelity of action potential transmission. Indeed, this is presumably the origin of perceived light flashes – known as phosphenes - in response to mechanical stimulation of the visual system, including nerve fiber tracts that connect the eye to the visual cortex (*Newton, 1669*; *Grüsser and Hagner, 1990*), or of the perception of sensory input along the forearm and small and ring fingers when the ulnar nerve is mechanically compressed as it courses superficially across the medial aspect of the elbow (*Clarke et al., 2007*). The mechanosensitive properties of TRAAK might function to counteract mechanical-induced depolarization in nodes by opening and hyperpolarizing the nodal membrane to reduce the probability of aberrant action potential initiation. Thus, mechanosensitivity of TRAAK channels could function as a molecular scale mechanism to protect against mechanically-induced action potentials that would function in concert with macroscopic scale protective mechanisms such as fiber pleating inside sheathed bundles.

Third, TRAAK gating could possibly respond to a mechanical component of the action potential. While this idea might seem far-fetched, mechanical displacements in nerves associated with action potentials have been documented experimentally (*Iwasa et al., 1980*; *Kim et al., 2007*). Optical and piezoelectric measurements showed swelling of crab nerve fibers during the action potential associated with an outward displacement of the nerve membrane (*Iwasa et al., 1980*). And high bandwidth atomic force microscopy showed that nerve terminals in the mammalian neurohypophysis undergo transient swelling associated with $Na^+$ influx during the action potential (*Kim et al., 2007*). While the physical origins of these mechanical transients remain uncertain, several theories have been considered, including swelling secondary to water and $Na^+$ entry through $Na^+$ channels (*Kim et al., 2007*), mechanical forces associated with $Ca^{2+}$ ion entry (*Rvachev, 2010*), and mechanical forces that are thermodynamically coupled to charging the capacitance of the membrane (*Hady El and Machta, 2015*; *Mosgaard et al., 2015*). Further work is needed to test the idea that mammals could make use of mechano-electric phenomena to activate TRAAK $K^+$ channels in the node of Ranvier.

## Materials and methods

### Expression and purification of mouse TRAAK

A gene encoding *Mus musculus* TRAAK (UniProt O88454) was codon-optimized for yeast expression, synthesized (Genewiz, Inc) and cloned into a modified pPICZ-B vector (Life Technologies Inc) to create a human rhinovirus 3C protease-cleavable C-terminal EGFP-10x histidine fusion protein construct. The predicted disordered C-terminal 123 amino acids were removed by PCR and two N-linked glycosylation sites (N81Q and N84Q) were removed by mutation. However, this construct was not expressed at sufficient levels for further study. Adding a sequence encoding amino acids 1–26 from *H. sapiens* TRAAK (MTTAPQEPPARPLQAGSGAGPAPGRA) to the N-terminus end of the *M. musculus* TRAAK gene improved expression yield ~8 fold. The resulting fusion protein *H. sapiens* TRAAK$_{1-26}$ – *M. musculus* TRAAK$_{1-275(N81Q,N84Q)}$-SNS-LEVLFQ/GP-EGFP-H10 was used for all experiments and is referred to as TRAAK in the text for simplicity.

This mouse TRAAK construct was transformed into *P. pastoris* SMD1163 and expressed via methanol induction in a fermenter (Infors HT, Inc) as described for human TRAAK (*Brohawn et al., 2012*). Frozen *Pichia* cells were disrupted by milling (MM301, Retsch Inc) 5 times for 3 min at 25 Hz. All subsequent purification steps were carried out at 4° C. Cell powder was added to lysis buffer (50 mM Tris pH 8.0, 150 mM KCl, 60 mM dodecyl-β-D-maltoside (DDM, Anatrace Inc), 100 μg/ml DNAse 1, 1 μg/ml pepstatin, 1 μg/ml leupeptin, 1 μg/ml aprotinin, 10 μg/ml soy trypsin inhibitor, 1 mM benzamidine, and 100 μg/ml AEBSF with 1 mM phenylmethylsulfonyl fluoride added immediately before use) at a ratio of 1 g cell pellet/4 mL lysis buffer. Membranes were extracted for 3 hr with gentle stirring followed by centrifugation at 35000 xg for 45 min. Cobalt resin (Clontech Inc) was added to the

supernatant (1 mL resin/5 g cell pellet) and stirred gently for 3 hr. Resin was collected on a glass column and serially washed and eluted in IMAC buffer (50 mM Tris pH 8.0, 150 mM KCl, 6 mM DDM) with 10 mM, 30 mM, and 300 mM imidazole pH 8.0. EDTA pH 8.0 (1 mM final) and purified human rhinovirus 3C protease (~1:50 wt:wt) were added to the eluted protein fraction and incubated with gentle rocking overnight. Cleaved protein was concentrated (50 kDa MWCO) and run on a Superdex 200 10/300 column (GE Healthcare) in SEC buffer (20 mM Tris pH 8.0, 150 mM KCl, 1 mM EDTA, 1 mM DDM). Peak fractions containing pure TRAAK were pooled and concentrated (50 kDa MWCO) for subsequent procedures.

## Generation and selection of anti-mouse TRAAK monoclonal antibodies

Monoclonal antibodies were raised in Armenian hamsters (*C. migratorius*) against mouse TRAAK purified in dodecyl-β-D-maltopyranoside (DDM) using standard procedures (Antibody and Bioresource Core Facility, The Rockefeller University) (*Brohawn et al., 2013*). Hybridoma cell lines were generated by fusion of hamster B cells to mouse myeloma cells. This cell line has not been authenticated or tested for mycoplasma contamination. Hybridoma supernatents were first screened for TRAAK binding by ELISA and positive clones were further tested for stable antibody-TRAAK channel complex formation by fluorescence-detection size exclusion chromatography (*Kawate and Gouaux, 2006*). Hybridoma supernatants (75 μL) were added to purified uncut TRAAK-EGFP (75 μL at 200 ng/uL in SEC buffer) and incubated at 4° C for 10 min. 100 μL of this reaction was injected on a Superdex 200 column run in SEC buffer. Antibodies that shifted TRAAK-EGFP fluorescence retention time to an earlier-eluting peak were purified for further analysis by immunostaining.

Antibodies were purified from the media supernatant (~100 mL) of hybridomas grown in disposable bioreactors (BD CELLine). Supernatant was neutralized by addition of 1:10 vol:vol 1M Tris pH 8.0 and applied to a 5 mL protein A Hitrap FF column (GE Healthcare Life Sciences) equilibrated in buffer A (100 mM Tris, 3 M NaCl, 10 mM EDTA pH 8.9). Antibody was eluted in Buffer B (10 mM sodium acetate, 150 mM NaCl pH 3), neutralized by addition of 1:10 vol:vol 1 M Tris 8, concentrated (30,000 MWCO) and applied to a Superpose 6 10/300 gel filtration column (GE Healthcare Life Sciences) run in storage buffer (10 mM Hepes, 135 mM NaCl, 15 mM KCl, 3 mM MgCl$_2$, 1 mM CaCl$_2$ pH 7.3). Peak fractions containing pure antibody were pooled, concentrated to ~3 mg/mL, snap-frozen in liquid nitrogen and stored at −80° C.

Antibodies were screened for TRAAK-selective labeling by immunostaining CHO-K1 cells expressing mouse K2P channels. Genes for full-length *M. musculus* K2Ps TRAAK (Uniprot O88454), TREK1 (P97438), TREK2 (Q8BUW1), TWIK1 (O08581), TWIK2 (Q3TBV4), TALK1 (G5E845), TASK2 (Q9JK62), TASK1 (O35111), TASK3 (Q3LS21), TASK5 (B2RVL1), THIK1 (Q8R1P5), THIK2 (Q76M80) and TRESK (Q6VV64) were codon optimized for eukaryotic expression, synthesized (Genewiz), and cloned into the EcoR1/Xho1 sites of a modified pCEH vector to generate C-terminally EGFP-tagged constructs (K2P-SNSAV- DAGLVPRGSAAA-EGFP-10x histidine) for transient transfection. Cells were grown in DMEM-F12 (Gibco) with 10% FBS, 2 mM L-glutamine, 100 units/mL penicillin and 100 ug/mL streptomycin in 96 well plates (Falcon 353072) and transfected with individual K2Ps using Fugene HD (Promega, Inc) according to the manufacturer's instructions. Two days after transfection, cells were fixed, permeabilized and labeled according to the following procedure. Fixation: wash twice with 100 uL media, fix with 200 uL 4% paraformaldehyde (vol/vol) in media 10 min at room temperature and wash twice with 100 uL media. Permeabilization: wash twice with 100 uL media, fix with 100 uL 0.15% Triton X-100 (vol/vol) in media 5 min at room temperature and wash twice with 100 uL media. Immunolabeling: wash twice with 100 uL media, bind primary antibody at 20 ug/mL in media 1 hr at room temperature, wash five times with 100 uL media, bind secondary antibody 1:500 (vol/vol) in media 1 hr at room temperature, wash three times with 100 uL media, wash twice in 100 uL DPBS and label DNA with 100 uL DPBS + nucblue (eight drops/10 mL). Fixation and permeabilization were omitted and all steps were carried at 37° C and 5% CO$_2$ for live cell labeling. Cells were imaged on a Molecular Devices ImageXpress Micro XL with DAPI, FITC and Texas Red filters.

Polyclonal antibodies were raised in rabbits against mouse TRAAK purified in dodecyl-β-D-maltopyranoside (DDM) (Covance, Inc) as described above. Polyclonal antibody 2134 was assessed for selective labeling of TRAAK among mouse K2Ps as described above and found to be specific at dilutions up to 1:500. At this concentration, it also labels rat TRAAK heterologously expressed in HEK293T cells (Figure 5-figure supplement 3) and was therefore used for immunostaining of TRAAK

in rat sciatic nerve. Polyclonal antibody 2134 supernatent was purified using a Protein A affinity column (GE Healthcare). Serum was neutralized with 1/10 vol 1 M Tris pH 8, bound to the column equilibrated in 100 mM Tris, 3 M NaCl, 10 mM EDTA, pH 8.9, and eluted with 10 mM NaAcetate, 150 mM NaCl, pH 3. Fractions containing antibody were neutralized with 1/10 vol Tris pH 8.0, concentrated, and further purified on a Superpose 6 10/300 gel filtration column (GE Healthcare Life Sciences) run in storage buffer (10 mM Hepes, 135 mM NaCl, 15 mM KCl, 3 mM MgCl$_2$, 1 mM CaCl$_2$ pH 7.3). Peak fractions containing pure antibody were pooled, frozen in liquid nitrogen, and stored at −80˚ C.

## mTRAAK-Fab complex preparation, crystallization and structure determination

Purified antibody clone 1B10 at ~1.5–2 mg/mL in 100 mM Tris, 10 mM sodium acetate, 150 mM NaCl pH 8.0 was mixed with 1:10 vol:vol 100 mM betamercaptoethanol, 100 mM EDTA, 100 mM L-cysteine HCl pH 8 and 1:100 1 mg/mL papain in PBS and incubated 2 hr at 37˚ C to generate Fab fragments. The digest was stopped by addition of 1:10 vol:vol 230 mM iodoacetamide and dialyzed against 10 mM Tris pH 8.0 10 mM KCl overnight. Fab fragments were separated from constant regions and uncut antibody on a 5 mL Hitrap Q sepharose FF anion exchange column over a salt gradient from 10 mM Tris pH 8.0 10 mM KCl to 10 mM Tris pH 8.0 1M KCl. Pure Fab fractions were pooled and concentrated to ~360 uM for complex formation. mTRAAK was purified as described above except n-decyl-β-D-maltopyranoside(DM) was used in place of dodecylmaltoside. Purified mTRAAK in 20 mM Tris pH 8.0 150 KCl 1 mM EDTA 4 mM DM was mixed with 1.25 molar excess Fab at 4˚ C, incubated for 10 min and run on a Superdex 200 10/300 column (GE Healthcare Life Sciences) equilibrated in 20 mM Tris pH 8.0 150 KCl 1 mM EDTA 4 mM DM. Pure complex fractions were pooled and concentrated to 30 mg/mL for crystallization.

Crystals were grown by hanging drop vapor-diffusion in drops of 0.15–0.25 uL protein added to an equal volume of reservoir solution consisting of 50 mM Sodium Citrate pH 5.5, 500 mM KCl and 9–13% (w/v) PEG4000. For cryoprotection, crystals were successively transferred through drops of mother liquor with increasing PEG4000 up to ~30% (w/v) before being plunge frozen in liquid nitrogen.

Data were collected at beamline 24-IDE at the APS. Data from two crystals were processed with XDS, scaled without merging symmetry related reflections in XSCALE (*Kabsch, 2010*), and anisotropically truncated using the STARANISO server. Anisotropically truncated data were used for all steps of structure solution and refinement (elliptically fitted cutoffs of 3.877, 2.774, and 4.382 Å resolution along a,b, and c axes, respectively) as using these data improved map quality and interpretability compared to spherically truncated data. The structure was solved by molecular replacement in Phaser (*McCoy et al., 2007*) using a hamster Fab fragment (PDB 3LD8) with complementarity-determining loops removed and a poly-alanine model of human TRAAK with mobile TM4 and TM2-3 elements removed (PDB 4WFE). The structure was modelled in Coot (*Emsley et al., 2010*) and refined in Refmac (*Murshudov et al., 2011*) using local non-crystallographic symmetry restraints. Geometry of the model was assessed in Molprobity (*Chen et al., 2010*). The final model contains 1344 of 1490 amino acids (90%). Unmodeled amino acids in TRAAK include the termini (amino acids 1–26 from human TRAAK, amino acids 1, 259–275 and a nine amino acid linker including the protease cut site in mouse TRAAK), a loop connecting the helical cap to pore helix 1 (amino acids 102–111 or 112), and a loop connecting the selectivity filter to TM3 (amino acids 248–255 or 258) in each protomer. In the Fabs, unmodeled regions include a loop in each heavy chain (amino acids 137–145 or 146) and light chain (amino acids 45–50). Antibody-TRAAK interface analysis was carried out in PISA (*Krissinel and Henrick, 2007*). We note that the two antibody binding sites are slightly different from one another, presumably due a tilting of the helical cap away from the channel ionic conduction axis that is required for crystal contacts between asymmetric units.

## Planar lipid bilayer recording

Mouse TRAAK at 1 mg/mL was mixed with 10 mg/mL lipids (3:1 wt:wt 1-palmitoyl-2- oleoyl-sn-glycero-3-phosphoethanolamine: 1-palmitoyl-2-oleoyl-sn-glycero-3-phospho-(1'-rac-glycerol)) at a one protein:100 lipid wt:wt ratio in dialysis buffer (20 mM Hepes pH 7.4, 150 mM KCl, 1 mM EDTA) with 10 mM DDM. Detergent was removed by dialysis for 1 week at 4˚ C with biobeads added for the

final three days. Proteoliposomes were snap-frozen in liquid nitrogen and stored at −80° C. 20 mg/ml of 2:1:1 (wt:wt:wt) of 1,2-dioleoyl-sn-glycero-3-phosphoethanolamine (DOPE): 1-palmitoyl-2-oleoyl-sn-glycero-3-phosphocholine (POPC): 1-palmitoyl-2-oleoyl-sn-glycero-3-phospho-L-serine (POPS) was painted over a ~200 µm polystyrene hole separating two chambers. Vesicles were added to the *cis* chamber filled with 4 mL 10 mM Hepes pH 7.5, 150 mM KCl while the *trans* side contained 3 mL 10 mM Hepes pH 7.5, 15 mM NaCl. Once vesicles were fused with the bilayer NaCl was made 150 mM on the *trans* side. Voltage between the chambers was clamped with an Axopatch 200B amplifier in whole cell mode and current was filtered at 1 kHz, digitized at 10 kHz with a DigiData 1440A and recorded with Clampex software (Molecular Devices Inc).

## Tissue preparation, immunostaining, and microscopy

Adult (≥3 month) C57BL6 mice (wild-type or TRAAK KO, a gift from Florian Lesage and Michel Lazdunski) were used for all labeling experiments. Anesthetized mice were transcardially perfused with ice cold PBS followed by ice cold 4% paraformaldehyde in PBS. Tissue was harvested and post fixed in 4% PFA 30 min-overnight and subsequently washed three times in PBS for one hour each. Tissue was cryoprotected by successive exchanges into PBS with 5%, 15%, and 30% sucrose with equilibration judged by tissue sinking. Tissue was set in a mold with OCT, frozen in liquid nitrogen-cooled isopentane, and sectioned (16–20 um) onto slides and stored at −80°C until use. Sciatic nerves were harvested directly from euthanized mice without prior perfusion and fixed in 4% PFA in PBS prior to cryoprotection, freezing, and sectioning or were first teased onto glass slides. Whole mount skin preparations for staining of peripheral nerves, hair innervation, and sensory endings were prepared as described (*Chang et al., 2014*). Spinal cord was harvested by ejection or laminectomy without prior PFA perfusion (*Kennedy et al., 2013*). iDISCO was performed as described (*Renier et al., 2016*).

Tissue was stained as follows: slides were dried ≥30 min at 20°C, washed once with PBS at 4°C, permeabilized with PBST (PBS with 0.2% v/v Triton X-100) 15 min at 4°C, washed with PBS at 4°C, blocked with Image IT-FX Enhancer for 30 min at 20°C, washed with PBS at 4°C, washed with PBST at 4°C, blocked with PBST-NGS (PBST with 2% normal goat serum, Jackson ImmunoResearch Laboratories, Inc) for 1 hr at 20°C, incubated with primary antibodies in PBST at 4°C overnight, washed three times with PBST at 4°C, incubated with secondary antibodies in PBST for one hour at 20°C, washed twice with PBST at 4°C, washed twice with PBS at 4°C, mounted with Prolong glass with Nucblue (Invitrogen), dried for 24 hr, sealed, and stored at 4°C until imaged. Imaging was performed on a Zeiss LSM 780 laser scanning confocal microscope. Image analysis was performed in Fiji (*Schindelin et al., 2012*).

## Immunostaining of rat sciatic nerve fibers and rat TRAAK in HEK293T cells

Adult Wistar rats (500-700 g) were euthanized using $CO_2$ and the sciatic nerves were dissected. After removal of the epineurium and perineurium, fibers were teased apart using Dumont 5 fine tweezers in HEPES rat Ringer's solution (5 mM HEPES-Na pH7.2, 135 mM NaCl, 5.4 mM KCl, 1 mM $MgCl_2$ and 1.8 mM $CaCl_2$). Isolated fibers were fixed in rat Ringer's solution containing 4% paraformaldehyde at room temperature for 10 min, washed in phosphate-buffered saline (PBS, 10 mM $Na_2HPO_4$, 1.8 mM $KH_2PO_4$ pH 7.4, 137 mM NaCl, 2.7 mM KCl) and incubated with buffer A (PBS supplemented with 0.5% Triton-X 100% and 5% normal goat serum) for 4 hr with gentle agitation. Primary antibodies were diluted in solution A and incubated with fibers overnight at 4°C. CF-488A conjugated goat anti-mouse (Biotium 20018) and CF-555 conjugated goat anti-rabbit (Biotium 20232) antibodies were subsequently used for fluorescent labeling. Full-length *Rattus norvegicus* TRAAK was cloned between the EcoRI/XhoI restriction sites on a modified PCEH vector, resulting in a C-terminal GFP fusion. HEK293T cells were transfected with rat TRAAK using Lipofectamine 3000 Reagent (Invitrogen) according to manufacturer's protocols. 24 hr after transfection, the cells were fixed and stained as described for rat sciatic nerve, except that only Rabbit anti-TRAAK polyclonal antibody 2134 and the CF-555 conjugated goat anti-rabbit antibodies were applied. Epi-fluorescence and bright field images of stained fibers were taken on a Nikon Eclipse Ti inverted microscope using a Plan Apo 20 × 0.75 NA dry objective.

# Whole-cell recording of rat TRAAK heterologously expressed in CHO cells

CHO-K1 cells (ATCC, cultured as described above) were transfected with rat TRAAK-GFP with FugeneHD (Promega) following the manufacturer's instructions. Room-temperature whole-cell recordings were performed ~24 hr after transfection. Bath solution was rat Ringer's solution and pipette solution was rat axoplasmic solution and contained (in mM): 10 Tris-Cl pH 7.4, 160 KCl. To test the effect of RU2, 20 µM RU2 was supplemented to the bath solution and locally perfused at the cell. Borosilicate glass pipettes with resistance between 2 and 5 MΩ were used for room temperature whole-cell recordings. An Axopatch 200B amplifier connected to a Digidata 1440A digitizer (Molecular Devices) was used for data acquisition. Analog signals were filtered at 1 kHz and sampled at 20 kHz. A voltage protocol of holding at −80 mV, stepping to −100 mV to 60 mV for 200 milliseconds in 10 mV steps, then stepping back to −40 mV was used to record TRAAK current.

# Voltage clamp recordings from isolated rat nodes of Ranvier

The dissection of single myelinated nerve fibers followed the procedure as described (*Stämpfli and Hille, 1976*; *Nonner, 1969*). Adult Wistar rats (500-700 g) were decapitated during deep anesthesia induced by exposure to $CO_2$ and sciatic nerves were isolated. Following removal of the perineural sheaths from the sciatic nerve, a single myelinated nerve fiber was isolated over a distance of two internodes using a pair of fine forceps (Dumont no. 5) and angled spring scissors (FST, No.15010).

A schematic drawing of voltage clamping the node of Ranvier is shown in *Figure 5—figure supplement 1* (*Nonner, 1969*). The isolated nerve fiber was placed on top of the separating walls of the recording chamber filled with Ringer's solution (see *Figure 5—figure supplement 1A,B*) and fixed to the ridges with thin Vaseline seals. In addition to the electrical resistance provided by the Vaseline seals, the external resistance between pool B (connected to ground potential) and pool C was further increased by introducing an air gap (pool F in *Figure 5—figure supplement 1B*). The side pools C and E of the experimental chamber were filled with axoplasmic solution and both internodes were cut, creating a direct electrical connection between the axoplasm of the fiber and electrodes in C and E. The following solutions were used: Ringer's solution; isotonic KCl: 160 KCl, 2.2 $CaCl_2$; axoplasmic solution: 160 KCl. The pH was adjusted to 7.4 with Tris-HCl for all buffers. Chemicals except RU2 were purchased from Sigma. Measurements were performed at room temperature (20–23°C) or near physiological temperature (32–37°C).

The operating principle of the Nonner-clamp is as follows. The feedback amplifier keeps the potential at point C at ground potential by adjusting the potential at E. Any internodal current between C and D would also flow across the seal resistance between C and B and would produce a potential drop. The feedback amplifier prevented such a potential drop and thus eliminated flow of current between D and C. These two points were kept at ground potential. The potential at A with respect to ground corresponded to the absolute membrane potential of the node under investigation. It could be changed by means of the pulse generator. By passing current through the resistance between E and D, the external potential in A was forced to follow the command waveform generated by the pulse generator. The current flowing between E and D was identical to the membrane current of the node under investigation and was recorded as the voltage drop across the resistance of the internode ED assuming an internodal resistance of 15 MΩ (*Stämpfli and Hille, 1976*) (*Röper and Schwarz, 1989*). Membrane currents were low-pass filtered at 3 kHz. Capacity currents and leakage currents were not subtracted. Since the Nonner-clamp does not allow one to measure the absolute membrane potential, we used the following procedure to measure the membrane potential. Following successful mounting of a single nerve fiber in the recording chamber, we thus adjusted the holding potential to a value at which 30% of the Na channels were inactivated ($h_\infty$=0.7). This degree of Na inactivation was assumed to approximate the physiological holding potential. Previously (*Schwarz et al., 2006*), the absolute membrane potential ($E_H$) was determined by the following procedure: the node under investigation was superfused with isotonic KCl, and the amplitude of the potential step was determined to reach the reversal potential for $K^+$ ($E_K$) assuming $E_K$ = 0 mV. The value of $E_H$ measured in this way was −75 mV (*Schwarz et al., 2006*). A similar value for the resting potential was previously reported for rat nerve fibers: −80 mV (*Brismar, 1980*), −78 mV (*Neumcke and Stämpfli, 1982*) and −77 mV (*Röper and Schwarz, 1989*). A value of −70 mV in

the frog (*Stämpfli and Hille, 1976*) and −75 mV in the rat (*Schwarz et al., 2006*) were chosen in the present experiments.

Node recordings were made in Hamburg, Germany and in New York, USA. The dissection procedure and positioning of the single nerve fiber into the nerve chamber were identical in all recordings. Recordings in Hamburg used the amplifier as described by *Nonner (1969)*. For recordings in New York, a new custom-made amplifier was used as described below.

## Node-clamp using integrated operational amplifiers

The node-clamp for electrophysiological measurements of isolated nodes of Ranvier was originally reported in 1969 (*Nonner, 1969*). Since low impedance electrical access to the inside of the node is not possible, this method was designed, as described above, to keep the electrical potential of the inside of the node (point D, *Figure 5—figure supplement 1*) at the signal ground potential, 0 mV. This was achieved using a high bandwidth and very high input impedance amplifier built using discrete components. The original Nonner-clamp is not commercially available and very few are still in working order. For this reason, and since integrated operational amplifiers (opamps) have improved significantly over the past five decades, we re-built the Nonner's node-clamp using integrated opamps for experiments in New York.

The new node-clamp was based on an AD8033 operational amplifier (Analog Devices) which provides stable 80 MHz −3 dB gain-bandwidth product and over $10^{12}$ Ω input impedance (Figure S7). During construction, care was taken to minimize the stray capacitance between the output (point E) and the input (point C) of the main amplifier. Batteries were used as the power supply to avoid the 60 Hz sinusoidal contamination from the mains. Command voltages were generated using the Digidata 1440A digitizer (Molecular devices) controlled by the pClamp software. The output of the digitizer was combined with an offset voltage tunable with a potentiometer, scaled and buffered before application to point A. The output of the node-clamp was buffered, amplified and combined with an offset before conversion into digital signal using the digitizer. The current across the nodal membrane was calculated based on the estimated resistance between the node and one open end of the fiber (R9,∼15 MΩ) $I_{node} = V_E/R9$. In the current clamp configuration, the injection current was calculated in the same way from the command voltage applied at point E, $I_{inj} = V_E/R9$. Scaling factors were set in the 'lab bench' options in pClamp software to account for these conversions and the scaling factors in the circuit. This amplifier design enabled good recordings in the majority of isolated frog and rat nodes. Instabilities (ringing) were rare but occasionally observed at the apex of sodium current especially when the sodium current was very large and the fibers were thin. This circuit is versatile and could be easily improved in the future using more suitably developed opamps.

## Animals

All animal procedures were reviewed and approved by the Animal Care and Use Committee at The Rockefeller University, University of California, Berkeley or the University Hospital Hamburg-Eppendorf (Org 798/2016).

## Data availability

The X-ray crystallographic coordinates and structure factors for mouse TRAAK in complex with the 1B10 Fab are available at the Protein Data Bank under accession 6PIS.

## Acknowledgements

JRS wants to thank the Deutsche Forschungsgemeinschaft for financial support (Schw 292/17–1), Yvonne Pechmann for unfailing technical assistance, and Dr. Matthias Kneussel for continuous support. We thank staff at APS beamline 24-IDC/E, especially K Rajashankar, for assistance at the synchrotron, staff at the Rockefeller Bio-Imaging Resource Center for imaging support, N Renier, Z Wu, D Simon, and M Tessier-Lavigne for advice on iDISCO experiments, M Lazdunski and F Lesage for the TRAAK/TREK KO mice from which the TRAAK KO mouse line was generated, and Dr. F Sigworth for helpful discussions on designing of the opamp based node-clamp and N Hu for a discussion on conditional probabilities. RM is an Investigator in the Howard Hughes Medical Institute.

# Additional information

## Funding

| Funder | Author |
| --- | --- |
| Howard Hughes Medical Institute | Roderick MacKinnon |

The funders had no role in study design, data collection and interpretation, or the decision to submit the work for publication.

## Author contributions

Stephen G Brohawn, Conceptualization, Data curation, Formal analysis, Supervision, Validation, Investigation, Visualization, Methodology, Writing—original draft, Writing—review and editing; Weiwei Wang, Conceptualization, Data curation, Formal analysis, Validation, Investigation, Visualization, Methodology, Writing—review and editing; Annie Handler, Data curation, Formal analysis, Validation, Investigation, Visualization, Methodology; Ernest B Campbell, Investigation, Methodology; Jürgen R Schwarz, Conceptualization, Resources, Data curation, Formal analysis, Supervision, Validation, Investigation, Visualization, Methodology, Writing—review and editing; Roderick MacKinnon, Conceptualization, Formal analysis, Supervision, Funding acquisition, Validation, Investigation, Methodology, Writing—original draft, Project administration, Writing—review and editing

## Author ORCIDs

Stephen G Brohawn (ID) https://orcid.org/0000-0001-6768-3406
Jürgen R Schwarz (ID) https://orcid.org/0000-0002-4141-6972
Roderick MacKinnon (ID) https://orcid.org/0000-0001-7605-4679

## Ethics

Animal experimentation: All animal procedures were reviewed and approved by the Animal Care and Use Committee at The Rockefeller University (Protocol ID 16912), University of California, Berkeley (AUP-2016-09-9174) or the University Hospital Hamburg-Eppendorf (Org 798/2016).

## Decision letter and Author response

Decision letter https://doi.org/10.7554/eLife.50403.021
Author response https://doi.org/10.7554/eLife.50403.022

# Additional files

## Supplementary files

• Supplementary file 1. Data collection and refinement statistics.
DOI:
• Supplementary file 2. Antibodies used in this study.
DOI:
• Transparent reporting form DOI: https://doi.org/10.7554/eLife.50403.017

## Data availability

Coordinates and structure factors data have been deposited in PDB under the accession code 6PIS.

The following dataset was generated:

| Author(s) | Year | Dataset title | Dataset URL | Database and Identifier |
| --- | --- | --- | --- | --- |
| Brohawn SG | 2019 | Mouse two pore domain K+ channel TRAAK (K2P4.1) - Fab complex structure | https://www.rcsb.org/structure/6PIS | 6PIS, Protein Data Bank |

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
