## [Decision Letter]

Thank you for submitting your article "The mechanosensitive ion channel TRAAK is localized to the mammalian node of Ranvier" for consideration by *eLife*. Your article has been reviewed by three peer reviewers, and the evaluation has been overseen by a Reviewing Editor and Richard Aldrich as the Senior Editor. The following individuals involved in review of your submission have agreed to reveal their identity: Fred Sigworth (Reviewer #1); James S Trimmer (Reviewer #2).

The reviewers have discussed the reviews with one another and the Reviewing Editor has drafted this decision to help you prepare a revised submission.

Summary:

This is a very nice and comprehensive study of the localization and function of TRAAK mechanosensitive potassium channels in the mammalian nervous system. The study is an important advance because, at long last, it identifies one of the channels that carry the "leak" current that maintains the resting potential and terminates action potentials in myelinated neurons.

Using "node clamp" on nodes of Ranvier in peripheral nerve fibers, the authors argue that while frog nodes that depend on high-voltage activated Kv1 channels for spike repolarization in nodes, rat nodes depend primarily on TEA-insensitive potassium leak channels for nodal repolarization. With a novel compound, RU2, developed in a previous high-throughput screen and shown to block TRAAK in expression systems, they show that an RU2-sensitive current contributes both to the resting potential and to action potential repolarization

Essential revisions:

The most significant concern raised by the reviewers is about the specificity of RU2 and the conclusion that TRAAK is responsible for a significant portion of the repolarizing current in rat axons. Is there a way to learn whether other channels such as the K2P also blocked by RU2 are expressed and functional at nodes? Part of the concern here is the apparent difference in the voltage-dependence (and possibly time-dependence) of the nodal currents and the heterologously expressed TRAAK currents. How do you explain these differences?

The identity of the TEA-sensitive K current in frog nodes of Ranvier is a smaller conclusion of the paper, but to identify it clearly as a KV1 channel it would be good to use a more specific blocker, such as DTX.

All of the reviewers felt that there was too much speculation about the importance of TRAAK's mechanosensitivity, in the absence of experimental evidence. The Discussion should be rebalanced to consider more of the previous work on TRAAK and on nodal currents, and less on the mechanosensitivity (the latter should probably also be removed or modified in the Abstract).

Please also improve the images in the figures, which multiple reviewers found difficult to read and interpret, also address the other comments (below) from the reviewers.

*Reviewer #1:*

This is a very careful and spectacularly comprehensive study of the localization and function of TRAAK mechanosensitive potassium channels in the mammalian nervous system. The study is an important advance because, at long last, it identifies one of the channels that carry the "leak" current that maintains the resting potential and terminates action potentials in myelinated neurons.

At the end of the Discussion the hypothesis is raised that the mechanosensitivity of TRAAK could result in its activation due to mechanical correlates of action potentials. This is a bold but maybe far-fetched hypothesis based on very little evidence. My own take is that it is good to raise the hypothesis, but only in a tentative manner, unless there is additional experimental or quantitative theoretical evidence that can be marshaled in support. Even a crude estimate of the size of mechanical disturbances in the nodal membrane, for example deduced from the optical results or the known water transport by Na channels, would be helpful in convincing a skeptical reader. Or, if water diffusion within the axon is slow enough to allow it, might one expect to see a brief "overshoot" of TRAAK current resulting from residual mechanical stress following the termination of Na^+^ current?

*Reviewer #2:*

The manuscript by Brohawn et al., focuses on expression and function of the TRAAK leak potassium channel in myelinated axons. Ion channels that control action potential propagation in myelinated axons are crucial to normal nervous system function, and their dysfunction contributes to disease. The identification of a novel ion channel present at the node of Ranvier is an important contribution to our understanding of this important neuronal compartment. The authors use newly developed and well characterized antibodies to effectively define TRAAK as a nodal ion channel in mouse CNS and PNS. They employ direct nodal recordings to look at single action potentials and how they are impacted by blockers of Kv channels and of TRAAK. The paper is in general clearly written and the figures clear and effective at conveying the results, with the minor exceptions noted below. There is a lot of information in the Results section in addition to the concise reporting of the experimental results, and as detailed this also generally holds for the Discussion. Both should be shortened and revised to make it more about this study. In particular, readers would be better served by a Discussion that focused on the validity of the authors' experimental observations, and by the authors discussing their findings in light of other published work dealing with the same or closely related subjects.

*Reviewer #3:*

This study by Brohawn et al. describes the novel finding that the tandem-pore potassium channel, TRAAK, is present at nodes of Ranvier both in peripheral and central neurons. To show this, the authors generate and extensively validate a TRAAK antibody using immunolabeling in TRAAK KO mice. In addition, they generate a crystal structure of the TRAAK protein bound to the antibody FAB fragment, they present immunostaining results showing a lack of binding to similar K2P channels, and they show in expression systems that the antibody does not interfere with the function of the channel. These thorough controls are a necessary foundation for the results shown throughout the rest of the study. Using the validated antibody, the authors then test the distribution of TRAAK within cells and conclude that it is expressed at nodes on an all-or-none basis.

The second half the paper tests the function of TRAAK in repolarization of axonal spikes using "node clamp" in rat or frog myelinated fibers. As part of this set of experiments, the authors designed and constructed an updated version of the classic node clamp amplifier and a description of the circuit is provided. Using this technique, the authors argue that while frog nodes that depend on high-voltage activated Kv1 channels for spike repolarization in nodes, rat nodes depend primarily on TEA-insensitive potassium leak channels for nodal repolarization. To demonstrate this, they use a novel compound, RU2, developed using a high-throughput screen and shown to block TRAAK in expression systems. In rat sciatic nerve, RU2 inhibited ~20% of the leak current in the node of Ranvier. Examination of spiking showed that RU2 depolarized the resting membrane potential which was accompanied by a reduction in spike height. They show that this is likely due to inactivation of Na channels.

This study describes in interesting and novel finding that TRAAK channels are present at nodes and contributes to their function. The authors should be commended for the substantial effort dedicated here to validating the antibody and for including the updated node clamp design. While the authors were rigorous in their validation of the antibody, a weakness of the study is that physiology component of the study depends almost completely on a novel drug, RU2, the specificity of which is unclear.

• The specificity of RU2 for TRAAK channels in native neurons is unclear. Specifically, I-V plots for the RU2-blocked current in CHO cells show roughly linear increases in current with voltage (Figure 6—figure supplement 1E). On the other hand, the RU2-blocked current in rat node (Figure 6) is small at hyperpolarized voltages and increases more prominently at voltages above -40 mV. Given the authors' published work showing that RU2 is not specific for TRAAK and blocks other K2P channels, this raises the question of whether multiple channels contribute to the RU2-blocked current at the node. The authors should provide more a definitive/complementary proof that TRAAK channels make a significant contribution to step-evoked currents and action potential firing in nodes. For example, they could test whether the RU2-blocked component is also sensitive to lipids.

Relatedly, it is important to know whether the example IV plots in Figure 6 represent typical RU2-blocked currents in node clamp recordings. Additional IV plot examples or an average quantification would help in comparing RU2-blocked currents in node clamp recordings to RU2-blocked currents recorded in expression systems.

• In Figure 4, the authors conclude that TRAAK is expressed in an all-or-none fashion based only on a limited sample size. This conclusion is based on evidence from only five (5) axons. In fact, only three TRAAK positive axons were analyzed, each of which had at most three nodes of Ranvier. To make a convincing point here, the authors would need to add more data.

To more robustly evaluate whether subpopulations of TRAAK positive and negative cells exist, however, the authors may consider fluorescence in situ hybridization experiments (FISH) quantifying TRAAK RNA in cell bodies would be much more convincing if they have access to this technique.

• Regarding Figure 7, the authors mention that RU2-induced depolarization was not always observed potentially due to the methodological issues. How often was this true and what was the threshold for considering a response RU2-insensitive? Please quantify. Even though depolarization was not always observed, it is still possible that RU2-block of TRAAK may lead to an increase in the input resistance of the node. What was the effect of RU2 on the input resistance?

• Supplementary Figure 4 is confusing and impossible to interpret. Specifically, it is unclear where (or if) labeling for the TRAAK positive puncta are present. The authors provide little in the way of specific descriptions of brain region or labeling of landmarks. Please improve this image and its presentation or consider removing.

• The images shown Figure 2 and 3 would benefit by adding higher magnification images of nodes, similar to those provided in Figures 2A and 3C. For the reader, these high-resolution images are necessary as the colocalization of proteins is difficult to determine in the low magnification figures. For example, the images showing validation of antibody in TRAAK KO mice are also difficult to see and compare with individual nodes in control mice.

As a general remark, the small size of the images in this study make evaluation of the data difficult. For example, the arrows in Figures 2D and Figure 3—figure supplement 1B are barely visible.

• Figure 6 describes the block of RU2 but does not mention what approximate voltage is being discussed. Along those lines, the authors mentioned that 10 out of 64 nodes were insensitive to RU2. Where these cells included in the values mentioned in the figure legends?

• The authors speculate extensively throughout the paper (Abstract, Introduction and Discussion) on potential function of TRAAK as a mechanosensitive channel in the node. However, there are no experiments/data in this study that directly test mechanosensation. The authors may want to add an experiment testing this idea or consider rebalancing their discussion of this topic.

---

## [Author Response]

Essential revisions:The most significant concern raised by the reviewers is about the specificity of RU2 and the conclusion that TRAAK is responsible for a significant portion of the repolarizing current in rat axons. Is there a way to learn whether other channels such as the K2P also blocked by RU2 are expressed and functional at nodes? Part of the concern here is the apparent difference in the voltage-dependence (and possibly time-dependence) of the nodal currents and the heterologously expressed TRAAK currents. How do you explain these differences?

We have made the following changes to the paper. First, to address the important issue of RU2 specificity we added this paragraph to the Discussion (second paragraph):

“Our data do not exclude the possibility that other K2P channels are also present in the node of Ranvier. […] Their identification is not the focus of this study.”

Second, to address the issue of different time- and voltage-dependent gating of TRAAK in CHO cells and in node of Ranvier we have added the following explanation (to the penultimate paragraph of section ‘Measurements of TRAAK currents in the node of Ranvier’):

“In nodes of Ranvier recorded at 32-37^o^ C, 10 or 20 µM RU2 blocked about 20% of the outward K^+^ current (Figure 6A-D, n=16 RU2-sensitive nodes). […] The currents in CHO cells are consistent with high levels of TRAAK activation and those in nodes of Ranvier are consistent with sub-maximally activated TRAAK.”

The identity of the TEA-sensitive K current in frog nodes of Ranvier is a smaller conclusion of the paper, but to identify it clearly as a KV1 channel it would be good to use a more specific blocker, such as DTX.

This paper is about TRAAK channels in the mammalian node of Ranvier. We have changed Kv1 to TEA-sensitive Kv channels.

All of the reviewers felt that there was too much speculation about the importance of TRAAK's mechanosensitivity, in the absence of experimental evidence. The Discussion should be rebalanced to consider more of the previous work on TRAAK and on nodal currents, and less on the mechanosensitivity (the latter should probably also be removed or modified in the Abstract).

We have removed speculation from the Abstract and have rebalanced our discussion on this point to highlight past experimental demonstrations of mechanical deflections associated with action potentials. This is raised in the last paragraph of the Discussion:

“Third, TRAAK gating could possibly respond to a mechanical component of the action potential. […] Further work is needed to test the idea that mammals could make use of mechano-electric phenomena to activate TRAAK K^+^ channels in the node of Ranvier.”

Please also improve the images in the figures, which multiple reviewers found difficult to read and interpret, also address the other comments (below) from the reviewers.

Images have been improved.

Additional changes to the manuscript: To improve the paper we read the individual reviewer comments and modified the paper to address many of the points raised (that were not in the summary). Many small edits were made and one major addition addresses the all-or-none presence of TRAAK in the last paragraph of the section ‘Localization of TRAAK in the nervous system’:

“It is impossible from sectioned tissue to determine whether TRAAK-positive and TRAAK-negative nodes can both occur within a single nerve fiber. To query this point, we isolated five axons from sciatic nerve; four of these contained three nodes and one contained two (Figure 4A,B). We observe that all nodes within a fiber are either positive (axons 1-3) or negative (axons 4-5). If we assume that positive and negative nodes can exist in a single fiber with random occurrence then, given that 80% of nodes in the sciatic nerve are positive (Figure S4), the probability of not observing a ‘mixed’ fiber among the five dissected nerves is ((0.8)^3^ + (0.2) ^3)4^((0.8)^2^ + (0.2) ^2^), which is slightly less than 0.05. The occurrence of TRAAK in nodes is therefore unlikely to be random. The data are more consistent with there being two kinds of fibers, those with and those without TRAAK.”